# Towards Multiscale Graph-based Protein Learning with Geometric Secondary Structural Motifs

**Shih-Hsin Wang**[1], **Yuhao Huang**[1], **Taos Transue**[1] , **Justin Baker**[2],
**Jonathan Forstater**[3], **Thomas Strohmer**[3] **& Bao Wang**[1]*
[1]Department of Mathematics and Scientific Computing and Imaging (SCI) Institute
University of Utah, Salt Lake City, UT 84102, USA
[2]Department of Mathematics, UCLA, Los Angeles, CA 90095, USA
[3]Department of Mathematics, UC Davis, Davis, CA 95616, USA

## Abstract

Graph neural networks (GNNs) have emerged as powerful tools for learning protein structures by capturing spatial relationships at the residue level. However, existing GNN-based methods often face challenges in learning multiscale representations and modeling long-range dependencies efficiently. In this work, we propose an efficient multiscale graph-based learning framework tailored to proteins. Our proposed framework contains two crucial components: (1) It constructs a hierarchical graph representation comprising a collection of fine-grained subgraphs, each corresponding to a secondary structure motif (e.g., $\alpha$-helices, $\beta$-strands, loops), and a single coarse-grained graph that connects these motifs based on their spatial arrangement and relative orientation. (2) It employs two GNNs for feature learning: the first operates within individual secondary motifs to capture local interactions, and the second models higher-level structural relationships across motifs. Our modular framework allows a flexible choice of GNN in each stage. Theoretically, we show that our hierarchical framework preserves the desired maximal expressiveness, ensuring no loss of critical structural information. Empirically, we demonstrate that integrating baseline GNNs into our multiscale framework remarkably improves prediction accuracy and reduces computational cost across various benchmarks.

## 1 Introduction

Machine learning (ML) has transformed computational protein modeling over the past decade [20, 42]. The 2024 Nobel Prize in Chemistry, awarded for groundbreaking contributions to computational protein design, highlights the transformative advancements [1]. Among ML techniques, graph-based methods excel at learning to encode complex chemical interactions in three-dimensional space, effectively handling spatial information. For instance, prior works (cf. [14, 17]) represent atoms as nodes and leverage edges to capture their interactions, such as chemical bonds and hydrogen bonds. This edge-based encoding enables graph neural networks (GNNs) to effectively model critical chemical interactions, which are essential for understanding protein structure-function relationships.

However, representing proteins at the atomic level incurs significant computational costs due to the sheer size and

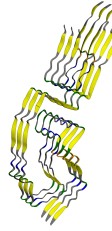

Figure 1: An example of two prion proteins with identical primary structures but distinct secondary structures. The normal form, hamster PrP$^C$ (left), contains $\alpha$-helical structures (marked in red). In contrast, its misfolded counterpart, PrP$^{Sc}$, on the right, lacks these helices and adopts a $\beta$-sheet-rich structure (marked in yellow). This structural change leads to abnormal aggregation, ultimately resulting in fatal consequences.

---

*Correspond to `wangbaonj@gmail.com`

39th Conference on Neural Information Processing Systems (NeurIPS 2025).

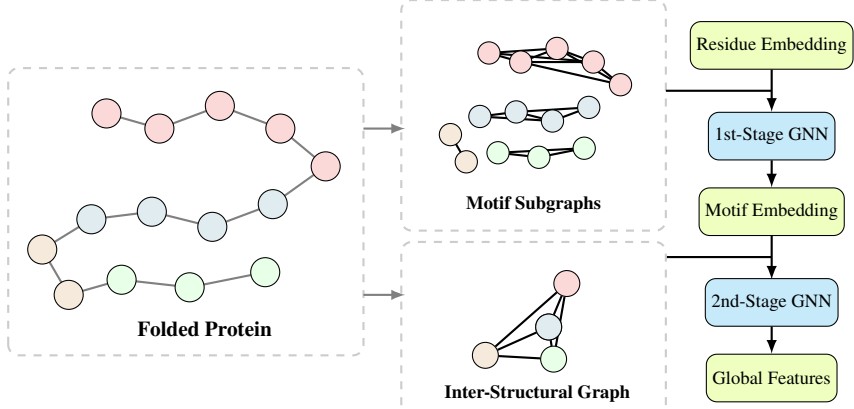

Figure 2: Overview of the proposed multiscale graph-based framework. We first construct a hierarchical graph representation that includes: (1) fine-grained motif subgraphs, where residues within each secondary structure motif (e.g., $\alpha$-helices, $\beta$-strands, loops) are treated as nodes, and (2) a coarse-grained structural graph, where each motif is abstracted as a single node. The first GNN operates independently on each motif subgraph to learn local embeddings. These learned motif-level features are then used to construct the coarse-grained graph, on which a second GNN performs message passing to model higher-level structure and generate the final prediction.

complexity of the resulting graphs. To address this, recent methods have shifted towards residue-level representations, where each residue serves as a single node in the graph (cf. [18, 48, 37, 41]). This coarse-graining not only reduces the graph size but also aligns naturally with the primary structure of proteins, where residues are the fundamental building blocks that determine protein properties and overall structure. However, existing residue-level approaches face challenges in capturing critical multiscale features. In particular, secondary structures—such as $\alpha$-helices and $\beta$-sheets—are formed by groups of residues and play a fundamental role in protein folding. Ignoring these higher-level structures can hinder the model's capacity to distinguish between biologically distinct protein states that share identical primary sequences. An illustrative example is the prion protein [6]. The normal cellular form of the prion protein, PrP$^{\text{C}}$, is typically found on the surface of healthy neurons. However, it can misfold into the pathogenic form, PrP$^{\text{Sc}}$, without any change in its primary structure. The key difference lies in the spatial arrangement of residues, which are reorganized into different types of secondary structures, resulting in a markedly altered folding pattern compared to the normal form (see Section 3 for details). This misfolded form aggregates abnormally due to its $\beta$-sheet-rich structures and induces pathogenic effects, ultimately leading to fatal neurodegenerative diseases. Figure 1 shows hamster PrP$^{\text{C}}$ (PDB: 1B10) and its misfolded form PrP$^{\text{Sc}}$ (PDB ID: 7LNA) [23, 24].

Some multiscale methods have been proposed to capture hierarchical features of protein structures. For example, recent approaches use distance-thresholded graphs with large radial cutoffs, combined with surface modeling, to encode multiscale information [33, 47]. While large-cutoff, distance-thresholded graphs can capture broader structural context, they often introduce significant computational overhead—both in runtime and memory footprint (see Section 5 for empirical evidence)—and may overlook biologically meaningful relationships identified by domain experts. These limitations underscore the *urgent need for a scalable framework that integrates domain knowledge to model intricate protein interactions efficiently while preserving expressive power*.

### 1.1 Our Contributions

In this paper, we propose a new multiscale GNN framework—depicted in Fig. 2 with detailed discussion in Sections 3 and 4—for learning proteins. Our key contributions are:

- We introduce a hierarchical, sparse, and geometry-aware graph representation of proteins by combining domain-expert algorithms to segment sequences into secondary structure motifs and constructing a multi-scale graph hierarchy: a collection of fine-grained graphs that capture residue-level interactions within each secondary structure unit, and a single coarse-grained graph that models the spatial arrangement and relative orientation among these units. This representation preserves geometric fidelity while providing a provable sparsity bound on the total number of edges, which is critical for scalability and efficiency. See Section 3 for details.

- We develop a two-stage framework—leveraging two off-the-shelf GNNs operating in tandem to learn multiscale protein features based on the proposed hierarchical graph representation. The-

oretically, we characterize the maximal expressiveness of our framework, showing its ability to maintain spatial fidelity during message passing across different levels. See Section 4 for details.

- Empirically, we demonstrate that our multiscale framework enables existing GNN architectures to simultaneously improve prediction accuracy and computational efficiency (in both runtime and memory footprint) across benchmark tasks. See details in Section 5.

### 1.2 Related Works

**Protein Representation Learning.** A variety of deep learning approaches have been developed to model protein structures and functions by learning effective representations. Some methods leverage the sequential nature of proteins and employ convolutional neural networks (CNNs) [15] or large language models (LLMs) [45, 36] to learn directly from the amino acid sequence. However, since protein function is closely tied to its 3D structure, many recent efforts have shifted toward structure-based approaches. These methods represent proteins as graphs and use GNNs to capture spatial relationships. Notable examples of methods that learn geometric and symmetry-aware representations include [18, 17, 48, 37, 26, 41]. In addition, hybrid models such as DeepFRI [12] combine GNNs with sequence-level features extracted from pretrained protein language models, while ProtGO [16] integrates GNNs with a Gene Ontology encoder, achieving strong results with large-scale architectures. In contrast, our work focuses on 3D structure-based modeling and introduces an efficient, theoretically grounded hierarchical design that learns protein representation with substantially smaller model sizes.

**Multiscale Graph-Based Models for Protein Representation Learning.** Several recent efforts [14, 33, 47, 31] have explored multiscale GNN to better capture both local and global structural patterns. The methods in [33, 47] construct large-radius radial graphs combined with surface modeling, while [31] learns residue-level clustering for hierarchical representations. However, such approaches often incur high computational costs or rely on data-driven clustering without explicit biological grounding. In contrast, [14] incorporates domain principles by applying hierarchical pooling based on different types of residue interactions to extract multiscale features. However, their final pooling stages simplify the backbone chain by clustering every two consecutive residues along the sequence, without explicitly leveraging secondary structure motifs. As a result, capturing long-range dependencies (LRDs) requires repeated pooling over many layers, leading to high computational costs. Our work shares the multiscale motivation but differs by introducing a biologically grounded hierarchical graph construction inspired by prior molecular motif-based approaches [46, 43]. Specifically, we use secondary structures as high-level motifs to build a two-level graph that captures both local geometry and long-range dependencies. This design offers provable sparsity and expressiveness guarantees while maintaining high computational efficiency.

### 1.3 Organization

We organize this paper as follows: We recap on message-passing GNNs and the standard framework to analyze their expressiveness power, together with other necessary background materials in Section 2. We present our new hierarchical graph representations for proteins and two-stage GNN architectures in Section 3 and Section 4, respectively. We numerically validate the accuracy and efficiency of our proposed approach in Section 5. Technical proofs and additional experimental details are provided in the appendix.

## 2 Background

In this section, we provide background materials on point clouds, geometric graphs, local frames, and message-passing GNNs and their expressiveness characterizations.

**Point Clouds and Geometric Graphs.** A *3D point cloud* is a collection of points in $\mathbb{R}^3$, i.e., $\{\boldsymbol{x}_i\} \subset \mathbb{R}^3$. Each point may have a *feature vector* $\boldsymbol{f}_i \in \mathbb{R}^d$, capturing additional attributes beyond the spatial coordinates. We denote such an attributed point cloud as $\{\boldsymbol{x}_i, \boldsymbol{f}_i\}_{i=1}^N$. Two point clouds $\{\boldsymbol{x}_i, \boldsymbol{f}_i\}_{i=1}^N$ and $\{\tilde{\boldsymbol{x}}_i, \tilde{\boldsymbol{f}}_i\}_{i=1}^N$ are considered *identical up to rigid motions* if there exists a bijection $\sigma : \{1, \ldots, N\} \to \{1, \ldots, N\}$ and a rigid motion $g$ such that $\boldsymbol{f}_{\sigma(i)} = \tilde{\boldsymbol{f}}_i$ and $\boldsymbol{x}_{\sigma(i)} = g \cdot \tilde{\boldsymbol{x}}_i$ for all $i$.

Extending this concept, a *geometric graph* $\mathcal{G} = (\mathcal{V}, \mathcal{E}, \boldsymbol{F})$ introduces a graph structure over the point cloud to model geometric relationships through edges. Here, $\mathcal{V}$ is the set of nodes, $\mathcal{E}$ the set of edges, and $\boldsymbol{F} = [\boldsymbol{f}_1, \ldots, \boldsymbol{f}_n]$ is the matrix of node features, which may also encode geometric attributes. When edges are equipped with features, $e_{ij} \in \mathcal{E}$ denotes both the edge and its associated attributes.

**Message Passing GNNs.** Consider a (geometric) graph $\mathcal{G} = (\mathcal{V}, \mathcal{E}, \boldsymbol{F})$. Starting from $\boldsymbol{f}_i^{(0)} = \boldsymbol{f}_i$, message passing GNNs propagate features from iteration $t$ to $t+1$ as follows:

$$\boldsymbol{f}_i^{(t+1)} = \text{UPD}\left(\boldsymbol{f}_i^{(t)}, \text{AGG}(\{\!\!\{ \boldsymbol{f}_i^{(t)}, \boldsymbol{f}_j^{(t)}, e_{ij} \mid j \in \mathcal{N}_i \}\!\!\})\right),$$
$$\boldsymbol{f} = \text{readout}(\{\!\!\{ \boldsymbol{f}_i^{(T)} \mid i \in \mathcal{V} \}\!\!\}),$$

(1)

where $e_{ij}$ represents the attribute of edge $(i, j)$, $\mathcal{N}_i$ denotes the neighborhood of node $i$, consisting of nodes in $\mathcal{V}$ directly connected to $i$ by an edge in $\mathcal{E}$, $\{\!\!\{ \cdot \}\!\!\}$ denotes a multiset, and UPD, AGG, and readout are learnable functions, parameterized by multilayer perceptions.

**Maximal Expressive GNNs.** The expressiveness of GNNs is often analyzed through the lens of the Weisfeiler-Lehman (WL) graph isomorphism test [44, 28], which provides a theoretical foundation for distinguishing non-isomorphic graphs. A GNN is said to be *maximally expressive* if its key components, i.e., UPD, AGG, and readout, are injective [19, 40]. This notion can be viewed as pushing a given GNN architecture to its theoretical expressive limit under ideal conditions. Under this assumption, we ask whether a given GNN architecture can distinguish all non-isomorphic graph structures; that is, whether a maximally expressive GNN can produce distinct readout features for non-isomorphic graphs, given a sufficient number of layers $T$.

This theoretical framework has been extended to geometric graphs, particularly those derived from point clouds, where node and edge features encode spatial or geometric information. A recent study [41] investigates whether maximally expressive GNNs can distinguish point clouds—up to rigid motions—by operating on their corresponding SCHull graphs (see Appendix C for a review), specific geometric graphs constructed from these point clouds. Since our framework adopts the SCHull graph as its underlying representation, we summarize the relevant expressiveness result below:

**Theorem 2.1.** *[41] Let $F$ be a maximally expressive GNN with depth $T = 1$. Then $F$ can distinguish between the attributed SCHull graphs of any two non-isomorphic generic point clouds.*

*Remark* 2.2. The genericity of point clouds refers to the condition that the point coordinates are algebraically independent over the field of rational numbers—i.e., they do not satisfy any nontrivial polynomial equation with rational coefficients. This condition holds for most protein structures encountered in practice. Therefore, we assume genericity for all structures studied in this work.

**Local Frames.** A *local frame* is an orthogonal matrix $g \in \text{O}(3)$, consisting of three orthonormal vectors that define a local 3D coordinate system. Notably, local frames are *equivariant* under rotations and reflections: when a rotation or reflection is applied to an object's coordinates, the associated local frame transforms accordingly. Formally, if $g(\boldsymbol{X})$ denotes the local frame computed from a matrix of coordinates $\boldsymbol{X}$, then for any $h \in \text{O}(3)$, we have $g(h \cdot \boldsymbol{X}) = h \cdot g(\boldsymbol{X})$. In molecular and structural modeling, spatial units such as functional groups are often associated with such local frames to capture their orientations [9, 8]. Given two spatial units with associated local frames $g_i$ and $g_j$, the product $g_i^\top g_j$ represents the rotation (or reflection) that aligns one frame with the other, thereby encoding their relative orientation [8]. This formulation provides a principled way to compare geometric configurations and serves a role analogous to transition maps in differential geometry, enabling accurate geometric information to pass across local reference systems.

## 3 Hierarchical Graph Representations for Proteins

In this section, we present a new multiscale hierarchical graph representation for proteins. We begin by discussing the rationale for hierarchical graph construction, designed to capture protein structures at multiple levels of granularity. Next, we describe the graph construction process, detailing how it represents structural information within and across secondary structures and preserves critical spatial features at each level.

### 3.1 Protein Hierarchical Structures

Proteins exhibit an inherently hierarchical organization, structured across multiple scales. At the most fundamental level, they can be considered as linear sequences of amino acids, known as the primary structure. Current trends in graph-based protein modeling leverage this residue-level information by representing each amino acid as a node, with edges capturing chemical bonds and spatial proximities. This graph-based representation effectively models local interactions, reflecting the biological principles underlying protein folding.

However, the primary structure is only the first step in the hierarchical organization of proteins. As folding progresses, *contiguous sequences* of residues self-assemble into more complex geometric mo-

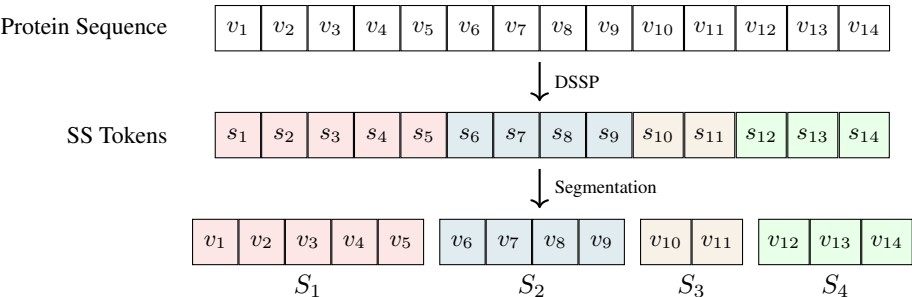

Figure 3: A visual illustration of the identification and segmentation process for protein secondary structures. Each residue $v_k$ is assigned a secondary structure (SS) token $s_k$ by DSSP, and consecutive residues with the same token are grouped into subsequences $S_i$.

tifs—such as $\alpha$-helices and $\beta$-strands—forming the secondary structure. These recurring geometric patterns are stabilized through hydrogen bonding and are essential not only for maintaining the overall structure of the protein but also for determining its functions and interactions with other molecular units. This multiscale complexity underscores the necessity for a hierarchical modeling approach that can effectively capture both local and global structural features, as well as LRDs inherent in protein folding.

## 3.2 Identification and Segmentation of Protein Secondary Structures

The identification of secondary structures is a critical step in protein modeling, providing insights into the folding patterns and spatial organization of amino acid sequences. Various methods have been developed for this purpose, especially the DSSP (Define Secondary Structure of Proteins) algorithm [21, 34] being one of the most widely used due to its robustness and accuracy. DSSP analyzes a protein's backbone conformation by first identifying backbone-backbone hydrogen bonds (H-bonds) based on geometric criteria and hydrogen-bond energy calculations. It then uses these H-bonds to detect structural motifs, including turns, bridges, $\alpha$-helices, and $\beta$-sheets. A detailed description of the DSSP algorithm is provided in Appendix D.

| Token | Secondary Structure |
|---|---|
| 'H' | $\alpha$-helix |
| 'B' | Isolated $\beta$-bridge |
| 'E' | Strand (all other $\beta$-ladder residues) |
| 'G' | $3_{10}$-helix |
| 'I' | $\pi$-helix |
| 'P' | $\kappa$-helix (poly-proline II helix) |
| 'T' | Turn |
| 'S' | Bend |
| '-' | None |

Table 1: Secondary structure tokens and their corresponding types.

For a protein with amino acid sequence represented as $\{v_k\}_{k=1}^N$, where each $v_k$ denotes a residue, DSSP assigns a secondary structure token $s_k$ to each residue, indicating its structural type (e.g., $\alpha$-helix, $\beta$-strand, or none). Table 1 summarizes the complete list of secondary structure tokens and their corresponding motifs. This process yields an annotated sequence $\{(v_k, s_k)\}_{k=1}^N$, where each residue is paired with its secondary structure token. We then use this annotation to segment the sequence into a set of structurally coherent subsequences, denoted by $\{S_i := \{v_k\}_{k=n_i}^{n_{i+1}-1}\}_{i=1}^I$. Each subsequence $S_i$ consists of consecutive residues that share the same token of the secondary structure $s_k$. Figure 3 illustrates the process of identifying and segmenting the protein sequence based on secondary structure annotations.

The resulting set of subsequences $\{S_i\}_{i=1}^I$, grouping residues by their secondary structure types, forms the foundation for constructing the hierarchical graphs. Each subsequence—representing a distinct secondary structural motif—will serve as a node in the higher-level graph representation.

## 3.3 Construction of Hierarchical Geometric Graphs

Building upon segmenting a protein sequence into secondary structure-based subsequences $\{S_i\}_{i=1}^I$, we construct a hierarchical geometric graph representation that captures both fine-grained and coarse-grained relationships. Specifically, we construct a collection of *intra-structural graphs*, each modeling residue-level interactions within a secondary structure unit by treating residues as nodes, and a single *inter-structural graph* that captures the spatial organization and relative orientation among these units, treating each unit as a node. This multiscale design enables the framework to preserve detailed geometric features within motifs while capturing LRDs across the overall protein structure. An illustrative example of our hierarchical graph construction is provided in Fig. 4. To ensure geometric completeness and computational efficiency, we adopt the SCHull graph construction method [41]

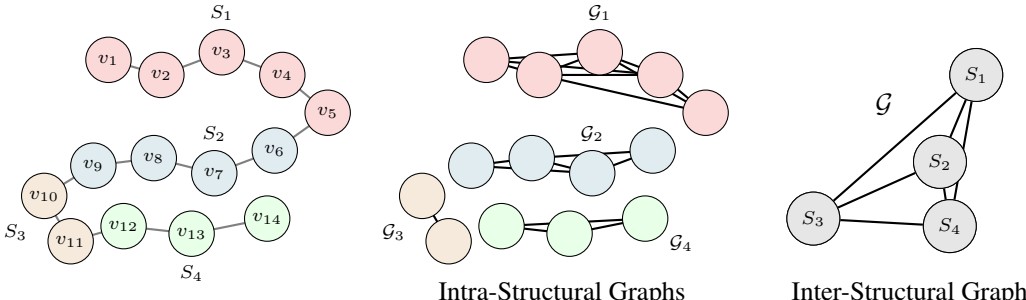

Figure 4: Hierarchical geometric graph construction. Left: A synthetic protein-like structure composed of 14 residues $\{v_k\}_{k=1}^{14}$, grouped into four secondary structure subsequences $\{S_i\}_{i=1}^4$. Middle: Intra-structural graphs $\mathcal{G}_i$ capture local information within each subsequence $S_i$ using SCHull. Right: The inter-structural graph $\mathcal{G}$ is formed by connecting the geometric centers of each $S_i$, modeling higher-level structural relationships between secondary structural motifs.

(see also Appendix C), which constructs sparse yet rigid graphs based on node coordinates. We now describe how this method is applied at both levels of our hierarchical construction:

**Intra-Structural Graph $\mathcal{G}_i$.** For each secondary structure unit $S_i$, we build an intra-structural graph $\mathcal{G}_i$, where nodes represent residues within $S_i$, and edges are computed based on the 3D coordinates of their $\alpha$-carbon atoms using the SCHull method. Node and edge features include geometric features generated by SCHull, combined with residue-specific attributes (e.g., amino acid type).

**Inter-Structural Graph $\mathcal{G}$.** The inter-structural graph $\mathcal{G}$ is built by treating each secondary structure unit $S_i$ as a node, with its coordinate defined by the geometric center of its residues. SCHull is again applied to determine edges between these structural units based on the spatial arrangement of their centers. In addition to geometric features on SCHull, we incorporate an additional edge feature, $g_i^\top g_j$, where $g_i = \mathcal{F}(\mathcal{G}_i)$ denotes a local frame computed from $\mathcal{G}_i$. The product $g_i^\top g_j$ captures the relative 3D orientation between two secondary structure units, following the orientation encoding proposed in [8] (see also Section 2 for details). This orientation encoding is the key to ensuring the expressiveness guarantee proved in Section 4. The construction of frames is discussed in Appendix B.

*Remark* 3.1. Dihedral angles are widely used to capture relative orientations between adjacent residues [20] or local structural motifs [11, 38]. In contrast, our approach leverages local frames to encode relative orientations between secondary structure units. Notably, as shown in [8], dihedral angles are inherently contained within the product $g_i^\top g_j$, which encodes richer geometric features.

Beyond geometric fidelity, our hierarchical construction also ensures strong sparsity, which is crucial for scaling GNNs to long protein sequences. The following proposition provides an upper bound on the total number of edges in the hierarchical graph:

**Proposition 3.2.** *Let $N$ be the total number of residues in a protein, denoted by $\{v_k\}_{k=1}^N$, and let $\{S_i\}_{i=1}^I$ represent its segmentation into secondary structure units. For each unit $S_i$, let $\mathcal{G}_i$ denote the intra-structural graph, and let $\mathcal{G}$ be the inter-structural graph connecting the structural units. Let $\mathcal{E}_i$ and $\mathcal{E}$ denote the sets of edges in $\mathcal{G}_i$ and $\mathcal{G}$, respectively. Then the total number of edges in the full hierarchical representation satisfies:*

$$|\mathcal{E}| + \sum_{i=1}^I |\mathcal{E}_i| < 3N.$$

In Section 5, we report the total number of edges created in our framework, along with the average runtime and memory usage, and compare these metrics against existing methods across benchmarks to highlight the efficiency gains enabled by our sparse hierarchical design.

## 4 A Two-Stage GNN Architecture for Multiscale Protein Modeling

Finally, we introduce a two-stage GNN framework that leverages our multiscale graph representation for efficient and expressive protein learning. Figure 2 illustrates the overall architecture.

The first-stage GNN operates independently on each intra-structural graph $\mathcal{G}_i$, where each graph corresponds to a single secondary structure unit. It encodes local geometric and chemical interactions among residues and generates embeddings that summarize each unit's internal feature. These embeddings are then passed to the second-stage GNN, which treats each secondary structure unit as a node in the inter-structural graph $\mathcal{G}$. This graph captures spatial and functional relationships between structural motifs, enabling the second GNN to model LRDs and global features of the protein. Figure 2 provides an overview of this multiscale learning framework.

**Message Passing within Secondary Structure Units.** The first-stage GNN applies message passing to each intra-structural graph $\mathcal{G}_i$, capturing local geometric and chemical interactions and producing a compressed embedding $\boldsymbol{s}_i$ for each secondary structure unit:

$$
\begin{aligned}
\boldsymbol{f}_k^{(t+1)} &= \mathrm{UPD}_1\left(\boldsymbol{f}_k^{(t)}, \mathrm{AGG}_1(\{\!\{\boldsymbol{f}_k^{(t)}, \boldsymbol{f}_l^{(t)}, \boldsymbol{e}_{kl} \mid l \in \mathcal{N}_k(\mathcal{G}_i)\}\!\})\right), \quad \text{for } t = 0, 1, \ldots, T_1 - 1, \\
\boldsymbol{s}_i &= \mathrm{readout}_1(\{\!\{\boldsymbol{f}_k^{(T_1)} \mid k \in \mathcal{V}(\mathcal{G}_i)\}\!\}),
\end{aligned}
\tag{2}
$$

where $\boldsymbol{f}_k^{(0)}$ denotes the initial node feature of residue $v_k$ (e.g., amino acid type) and $\boldsymbol{e}_{kl}$ denotes the attribute of edge $(k, l)$ on $\mathcal{G}_i$, $\mathcal{N}_k(\mathcal{G}_i)$ is the neighborhood of node $k$ on $\mathcal{G}_i$, $\mathrm{UPD}_1$ updates node features, $\mathrm{AGG}_1$ aggregates neighbor features, and $\mathrm{readout}_1$ produces the final embedding.

**Message Passing across Secondary Structure Units.** The second-stage GNN operates on the inter-structural graph $\mathcal{G}$, where each node represents a secondary structure unit. Node features are initialized using the embeddings produced by the first-stage GNN. This stage then performs message passing over $\mathcal{G}$ and ultimately outputs the global feature vector $\boldsymbol{s}_{\mathrm{global}}$, which serves as the final output of our framework:

$$
\begin{aligned}
\boldsymbol{s}_i^{(t+1)} &= \mathrm{UPD}_2\left(\boldsymbol{s}_i^{(t)}, \mathrm{AGG}_2\left(\{\!\{(\boldsymbol{s}_i^{(t)}, \boldsymbol{s}_j^{(t)}, \boldsymbol{e}_{ij}) \mid \mathcal{G}_j \in \mathcal{N}_{\mathcal{G}_i}\}\!\}\right)\right), \quad \text{for } t = 0, 1, \ldots, T_2 - 1, \\
\boldsymbol{s}_{\mathrm{global}} &= \mathrm{readout}_2(\{\!\{\boldsymbol{s}_i^{(T_2)} \mid i \in \mathcal{V}(\mathcal{G})\}\!\}),
\end{aligned}
\tag{3}
$$

where $\boldsymbol{s}_i^{(0)} = s_i$, $\boldsymbol{e}_{ij}$ denotes the attribute of edge $(i, j)$ on $\mathcal{G}$, $\boldsymbol{s}_{\mathrm{global}}$ represents the final output feature of our framework. The functions $\mathrm{UPD}_2$, $\mathrm{AGG}_2$, and $\mathrm{readout}_2$ denote the update, aggregation, and readout operations, respectively.

**Maximal Expressiveness of Two-Stage GNN Framework.** We now provide a theoretical characterization of the expressiveness of our proposed multiscale hierarchical learning framework. This analysis builds on the notion of maximal expressiveness introduced in Section 2 (see also Theorem 2.1), a standard approach for evaluating the expressiveness of GNNs [44, 28, 19]. To formalize maximal expressiveness in our setting, we begin with the following assumption:

**Assumption 4.1.** $\mathrm{UPD}_1$, $\mathrm{UPD}_2$, $\mathrm{AGG}_1$, $\mathrm{AGG}_2$, $\mathrm{readout}_1$, and $\mathrm{readout}_2$ are injective.

This assumption is commonly adopted in theoretical analyses of GNNs to characterize the *best possible* representational power of an architecture [44, 28, 19, 40]. Crucially, it serves as a theoretical tool rather than a requirement for practical implementations, illustrating what the model could achieve under ideal conditions. Under this framework, we can formally state the following result on the maximal expressiveness of our model:

**Theorem 4.2.** *Let $F$ denote the two-stage GNN architecture defined in Section 4, with depths $T_1, T_2 \geq 1$, and using the hierarchical graph construction described in Section 4. Under Assumption 4.1, $F$ can distinguish any pair of protein structures that are not identical under rigid motions.*

*Remark* 4.3. In practice, we do not strictly enforce the injectivity required by Assumption 4.1, instead relying on sufficiently expressive MLPs with ReLU activations. Even when using non-injective pooling operations (e.g., mean pooling), models integrated with the SSHG framework still demonstrate consistent performance improvements, as shown in Section 5. Incorporating more expressive or injective aggregation schemes remains a promising direction for future work.

## 5 Numerical Experiments

We evaluate the effectiveness and efficiency of our proposed Secondary Structure-based Hierarchical Graph (SSHG) learning framework on two benchmark protein modeling tasks: enzyme reaction

classification [15] and protein-ligand binding affinity (LBA) prediction [39, 27]. Our goal is to assess the following:

**Efficiency:** The hierarchical design of SSHG (Figure 2) enables integrated GNNs to operate on sparse graphs, leading to a significant reduction in training time.

**Efficacy:** Despite operating on sparser graphs, SSHG-based models achieve superior accuracy, consistently matching or even exceeding state-of-the-art (SOTA) models. Full model configurations and dataset statistics are provided in Appendix E.2 and E.1.

**Experiment Setup:** All models are implemented using PyTorch Geometric [10] and trained on NVIDIA RTX 3090 GPUs. To mitigate overfitting, we follow [37] and apply Gaussian noise (std = 0.1) and anisotropic scaling in the range $[0.9, 1.1]$ to the node coordinates in both the original graph framework and SSHG framework. Additionally, we randomly mask amino acid types and secondary structure types with probabilities of 0.1 or 0.2. We apply the SCHull graph construction method [41] to construct intra-structural and inter-structural graphs. Specific training setups, architectures, and hyperparameters for different tasks are available in Appendix E.

**Baseline and Metrics:** We integrate our SSHG framework with several backbone models, including GVP-GNN [18], ProNet-Backbone [37], and Mamba [13]; see Appendix E.3 for implementation details. Models enhanced with SSHG are denoted by appending "+SSHG" to the original model name (e.g., Mamba+SSHG, ProNet+SSHG). We compare these SSHG-augmented models against a range of established baselines, including GCN [22], IEConv [14], DWNN [26], GearNet [48], HoloProt [33], GVP-GNN [18], and ProNet-Backbone [37], across two benchmark tasks: enzyme reaction classification (React)[15] and protein-ligand binding affinity prediction (LBA)[39, 27]. Performance is evaluated using classification accuracy for EC reaction classification, and standard regression metrics, including root mean square error (RMSE), Pearson correlation, and Spearman correlation for LBA. To further highlight the efficiency and scalability of our framework, we also report additional metrics in the ablation study, including training time per epoch (s/epoch), memory usage, number of model parameters, and the average total number of edges in the graph representations used by the GNNs.

## 5.1 EC Reaction Classification

Enzymes, which catalyze biological reactions, are categorized using Enzyme Commission (EC) numbers based on the types of reactions they facilitate [29]. In this task, we evaluate the performance of our SSHG-based models—ProNet-SSHG and Mamba-SSHG—on enzyme reaction classification to demonstrate the benefits of incorporating secondary structure information and encoding geometric relationships within/across structural motifs. We follow the same dataset and experimental setup as in [37, 15]. Details of the dataset splits and training settings are provided in Appendix E.1. Notice that the baseline GVP-GNN in [18] uses a radius cutoff of 4.5 Å, achieving an accuracy of 65.5%, while we increase the cutoff to 10 Å, which improves accuracy to 68.5%

| Method | Test Acc | Ave.Time (s/epoch) | # params |
|---|---|---|---|
| GCN [22] | 66.5 | 186 | – |
| GCN+SSHG (**ours**) | 71.2 | 150 | – |
| IEConv [14] | 87.2 | – | 9.8M |
| DWNN [26] | 76.7 | – | – |
| GearNet [48] | 79.4 | – | – |
| HoloProt [33] | 78.9 | 300 | 1.4M |
| GVP-GNN [18] | 68.5±0.1 | 334 | 1.0M |
| GVP-GNN+SSHG (**ours**) | 73.6±0.1 | 236 | 1.0M |
| ProNet-Backbone [37] | 86.4±0.2 | 210 | 1.3M |
| ProNet+SSHG (**ours**) | 87.2±0.2 | **140** | 1.3M |
| Mamba [13] | 85.9±0.2 | 236 | – |
| Mamba+SSHG (**ours**) | **88.4**±0.3 | 157 | 1.5M |

Table 2: Results of protein reaction classification. Here, "Ave.Time" denotes the average time for training one epoch.

As shown in Table 2, the use of SSHG consistently improves performance across baseline models, including GCN [22], GVP-GNN [18], ProNet [37], and Mamba [13]. In particular, ProNet-SSHG significantly reduces training time compared to the original ProNet-Backbone, while matching the best-performing baseline (IEConv) with far fewer parameters (1.3M compared to 9.8M). Mamba-SSHG further improves prediction accuracy, highlighting the benefits of integrating sequence modeling into our hierarchical framework. This task confirms the advantage of SSHG in boosting both computational efficiency and classification performance. For baseline comparisons, we adopt results reported in prior works and omit training time when not provided in the original papers, except for models integrated with SSHG, which we re-evaluate using our setup to ensure fair comparison. Reported baseline results are consistent across all tasks.

| Method | RMSE ($\downarrow$) | Pearson ($\uparrow$) | Spearman ($\uparrow$) | Ave.Time (s/epoch) ($\downarrow$) |
|---|---|---|---|---|
| TAPE [32] | 1.890 | 0.338 | 0.286 | – |
| IEConv [14] | 1.554 | 0.414 | 0.428 | – |
| Holoprot-Full Surface [33] | 1.464 | 0.509 | 0.500 | 45 |
| GCN [22] | 1.925 | 0.322 | 0.287 | 28 |
| GCN+SSHG (**ours**) | 1.788 | 0.392 | 0.359 | 23 |
| GVP-GNN [18] | 1.529 | 0.441 | 0.432 | 49 |
| GVP-GNN + SSHG (**ours**) | 1.488 | 0.512 | 0.477 | 35 |
| ProNet-Backbone [37] | 1.458 | 0.546 | 0.550 | 32 |
| ProNet+ SSHG (**ours**) | 1.435±0.004 | 0.579±0.004 | 0.591±0.003 | **24** |
| Mamba [13] | 1.457 ± 0.004 | 0.565 ± 0.003 | 0.554 ± 0.004 | 27 |
| Mamba+SSHG (**ours**) | **1.399**±0.003 | **0.614**±0.003 | **0.610**±0.004 | 29 |

Table 3: Results of LBA prediction task. Here, "Ave.Time" denotes the average time for training one epoch.

## 5.2 Ligand Binding Affinity

We further demonstrate the effectiveness of our SSHG framework on the benchmark task of protein-ligand binding affinity (LBA) prediction. Accurate LBA prediction plays a critical role in drug discovery by guiding the selection of promising drug candidates and minimizing the need for costly and time-intensive experiments. We evaluate our models—GCN+SSHG, GVP-GNN+SSHG, ProNet-SSHG, and Mamba-SSHG—using the PDBbind dataset [39, 27], following the experimental protocol established by [18], which includes a 30% sequence identity threshold to assess model generalization to unseen proteins. More details on the dataset and experimental setup are provided in Appendix E.1. To quantify how geometric features and secondary structure information enhance the predictive capacity and generalization ability of GNNs, we evaluate model performance on the test set using standard regression metrics: RMSE, Pearson correlation, and Spearman correlation. As shown in Table 3, ProNet-SSHG outperforms all baseline models, including the best baseline[2], ProNet-Backbone, in terms of both predictive accuracy and computational efficiency. Mamba-SSHG further improves performance across all three metrics while maintaining competitive training speed. These results confirm the advantages of integrating SSHG into existing architectures, enabling both higher accuracy and improved scalability.

## 5.3 Ablation Studies

In this section, we conduct ablation studies to investigate the impact of key components in our SSHG framework. Specifically, we examine: (1) the performance and efficiency trade-offs between existing dense or sparse radial graphs versus our SSHG-based construction, (2) architectural variations in the two-stage GNN design within SSHG under comparable parameter budgets, (3) the role of the hierarchical strategy, (4) the contribution of geometric features $g_i^\top g_j$, and (5) the effect of incorporating secondary structure (SS) information. Tables 4 and 5 present results for the first two factors. Due to space constraints, analysis of the remaining components is deferred to Appendix E.5 (Table 8). All experiments are performed on the EC reaction classification task, with each model trained for 300 epochs using a batch size of 16. We report test accuracy alongside training efficiency metrics, including average time per epoch and peak memory usage.

Table 4 compares training efficiency and resource usage across different graph construction strategies. For baseline models like ProNet and GVP-GNN, increasing the radial cutoff improves accuracy but incurs substantial computational costs. Raising the cutoff from 4 to 16 increases the average number of edges from ~1K to ~15K, leading to much higher memory usage and training time. While denser graphs enhance expressiveness, they are less practical for large-scale applications. In contrast, SSHG-based models achieve equal or better accuracy with far fewer edges and significantly lower computational overhead. This efficiency stems from the hierarchical design, which decouples local and global interactions. On both ProNet and GVPGNN, SSHG attains up to a 2× speedup in training time and a 90% reduction in memory usage while still improving accuracy.

Table 5 evaluates the robustness of SSHG to architectural variations, specifically how parameters are distributed between the two GNN stages. All configurations perform well, showing the framework's flexibility. Notably, allocating more capacity to the first-stage GNN slightly improves accuracy, suggesting that richer local (residue-level) representations are more beneficial than a larger global inter-structural stage alone in our SSHG framework.

---

[2]For these protein tasks, Mamba is implemented in our work rather than using prior implementations.

| Model | +SSHG | Cutoff | Avg. Num Edges | Time (s/epoch)↓ | Mem (MiB)↓ | Test Acc (%)↑ |
|---|---|---|---|---|---|---|
| ProNet | ✗ | 4 | 1,034.5 | 138 | 1,290 | 78.1 |
| | ✗ | 6 | 4,755.2 | 165 | 7,760 | 82.1 |
| | ✗ | 8 | 8,013.9 | 185 | 9,580 | 85.6 |
| | ✗ | 10 | 11,316.8 | 210 | 14,548 | 86.4 |
| | ✗ | 16 | 14,881.1 | 247 | 17,768 | 87.0 |
| | ✓ | – | 1,593.3 | 140 | 1,818 | 87.2 |
| GVP-GNN | ✗ | 4 | 1,034.5 | 216 | 1,558 | 65.5 |
| | ✗ | 6 | 4,755.2 | 254 | 3,828 | 66.9 |
| | ✗ | 8 | 8,013.9 | 298 | 6,286 | 68.1 |
| | ✗ | 10 | 11,316.8 | 334 | 8,930 | 68.5 |
| | ✗ | 16 | 14,881.1 | 354 | 11,248 | 69.2 |
| | ✓ | – | 1,593.3 | 236 | 1,416 | 73.6 |

Table 4: **Efficiency comparison.** Training efficiency and accuracy of different GNNs with and without SSHG across varying cutoff radii. SSHG achieves higher accuracy while substantially reducing runtime and memory usage.

| | MPGNN1 # params | MPGNN2 # params | Ave.Time (s/epoch) | Mem(MiB) | Test Acc |
|---|---|---|---|---|---|
| ProNet+SSHG | 0.69M | 0.69M | 140 | 1818 | 87.2 |
| | 1.03M | 0.34M | 136 | 2656 | 87.4 |
| | 0.34M | 1.03M | 142 | 1720 | 87.1 |
| GVPGNN+SSHG | 0.53M | 0.53M | 236 | 1416 | 73.6 |
| | 0.79M | 0.27M | 232 | 1451 | 75.3 |
| | 0.27M | 0.79M | 228 | 1372 | 71.6 |

Table 5: **Architecture Comparison**: Two-stage GNNs with varying size ratios between the first- (MPGNN1) and second-stage (MPGNN2) graph networks.

# 6 Concluding Remarks

In summary, we propose a multiscale and scalable GNN-based framework for protein representation and learning by leveraging a hierarchical graph construction that aligns naturally with biological structures. By combining domain knowledge of secondary motifs with a multiscale graph design, our approach captures both fine-grained residue-level interactions and coarse-grained structural relationships through a collection of intra-structural graphs, each corresponding to a secondary structure motif, and a single inter-structural graph that encodes their spatial arrangement and relative orientation. Theoretically, we establish that our framework preserves maximal expressiveness, ensuring no loss of critical geometric information. Empirically, we demonstrate consistent improvements in both predictive accuracy and computational efficiency across standard benchmarks. These results highlight the potential of our method as a general and flexible foundation for protein-based learning tasks, opening up new avenues for integrating biological priors into geometric deep learning.

In future work, we plan to extend our investigation beyond the current experiments on enzyme classification and ligand-binding affinity prediction. We aim to evaluate the framework on additional tasks such as fold classification and protein–protein interaction prediction. Moreover, we will explore architectural enhancements through injective aggregation schemes, more expressive pooling mechanisms, alternative motif definitions, and integration with pretrained protein language models to further improve the framework's generality and performance.

**Societal Impacts:** Our paper presents a new efficient and accurate machine learning model for learning biomolecules, which can impact structural biology and life sciences. We do not see additional negative societal impact compared to existing approaches due to our work.

## Acknowledgement

This material is based on research sponsored by NSF grants DMS-2152762, DMS-2208361, DMS-2219956, DMS-2208356, and DMS-2436344, and DOE grants DE-SC0023490, DE-SC0025589, and DE-SC0025801. This work is also supported by NIH grant R01HL16351.

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

# Appendices

## A  Missing proofs

**Proposition 3.2.** *Let $N$ be the total number of residues in a protein, denoted by $\{v_k\}_{k=1}^{N}$, and let $\{S_i\}_{i=1}^{I}$ represent its segmentation into secondary structure units. For each unit $S_i$, let $\mathcal{G}_i$ denote the intra-structural graph, and let $\mathcal{G}$ be the inter-structural graph connecting the structural units. Let $\mathcal{E}_i$ and $\mathcal{E}$ denote the sets of edges in $\mathcal{G}_i$ and $\mathcal{G}$, respectively. Then the total number of edges in the full hierarchical representation satisfies:*

$$|\mathcal{E}| + \sum_{i=1}^{I} |\mathcal{E}_i| < 3N.$$

*Proof of Proposition 3.2.* Recall that for a point cloud with $m > 2$ points, the SCHull algorithm constructs a geometric graph with at most $3m - 6$ edges [41]. For the remaining cases, when $m = 1$, no edges can be formed, and when $m = 2$, there is exactly one edge connecting the two points.

Now, consider the following partition of $\{1, 2, \ldots, I\}$:

$$
\begin{aligned}
J_1 &= \big\{i \,\big|\, |S_i| = 1\big\}, \\
J_2 &= \big\{i \,\big|\, |S_i| = 2\big\}, \\
J_{\geq 3} &= \big\{i \,\big|\, |S_i| \geq 3\big\}.
\end{aligned}
\tag{4}
$$

According to the construction of $S_i$, we have

$$|J_1| + |J_2| + |J_{\geq 3}| = I \text{ and } |J_1| + 2|J_2| + \sum_{i \in J_{\geq 3}} |S_i| = N. \tag{5}$$

Then we have for each intra-structural graph $\mathcal{G}_i$, constructed over the residues in secondary structure unit $S_i$, the number of edges satisfies:

$$
|\mathcal{E}_i| \leq
\begin{cases}
0 & \text{if } i \in J_1 \\
1 & \text{if } i \in J_2 \\
3|S_i| - 6 & \text{if } i \in J_{\geq 3}
\end{cases}
\tag{6}
$$

Summing over all intra-structural graphs gives:

$$
\begin{aligned}
\sum_{i=1}^{I} |\mathcal{E}_i| &= \sum_{i \in J_1} |\mathcal{E}_i| + \sum_{i \in J_2} |\mathcal{E}_i| + \sum_{i \in J_{\geq 3}} |\mathcal{E}_i| \\
&= 0 \cdot |J_1| + 1 \cdot |J_2| + \sum_{i \in J_{\geq 3}} 3|S_i| - 6 \\
&= |J_2| + 3\big(N - |J_1| - 2|J_2|\big) - 6|J_{\geq 3}| \\
&\leq 3\big(N - |J_1| - |J_2| - |J_{\geq 3}|\big) \\
&\leq 3(N - I).
\end{aligned}
\tag{7}
$$

For the inter-structural graph $\mathcal{G}$, which connects the $I$ secondary structure units, SCHull yields:

$$|\mathcal{E}| \leq 3I - 6.$$

Combining both bounds:

$$|\mathcal{E}| + \sum_{i=1}^{I} |\mathcal{E}_i| \leq (3I - 6) + 3(N - I) = 3N - 6 < 3N.$$

Therefore, the total number of edges in the hierarchical representation is strictly less than $3N$, completing the proof. $\qquad\square$

Before proving Theorem 4.2, we first recall Theorem 2.1, which will be applied in the argument:

**Theorem 2.1.** *[41] Let F be a maximally expressive GNN with depth $T = 1$. Then F can distinguish between the attributed SCHull graphs of any two non-isomorphic generic point clouds.*

**Theorem 4.2.** *Let F denote the two-stage GNN architecture defined in Section 4, with depths $T_1, T_2 \geq 1$, and using the hierarchical graph construction described in Section 4. Under Assumption 4.1, F can distinguish any pair of protein structures that are not identical under rigid motions.*

*Proof of Theorem 4.2.* Suppose the model $F$ assigns identical outputs to the hierarchical graphs of two protein structures represented by the point clouds $(\boldsymbol{X}, \boldsymbol{F})$ and $(\boldsymbol{X}', \boldsymbol{F}')$. Let $\mathcal{G}, \mathcal{G}'$ denote their respective hierarchical graphs, and let $\mathcal{G}_i, \mathcal{G}'_j$ denote the intra-structural graphs corresponding to their secondary structure units. Denote the first- and second-stage GNNs in $F$ by $F_1$ and $F_2$, respectively.

By construction, the inter-structural graph is defined over the set of tuples $\{(F_1(\mathcal{G}_i), \boldsymbol{z}_i)\}_{i=1}^{I}$ and $\{(F_1(\mathcal{G}'_j), \boldsymbol{z}'_j)\}_{j=1}^{J}$, where $\boldsymbol{z}_i$ and $\boldsymbol{z}'_j$ are the geometric centers of the secondary structure units $\mathcal{G}_i$ and $\mathcal{G}'_j$, respectively. According to Theorem 2.1, $F_2$ assigns identical outputs to $\mathcal{G}$ and $\mathcal{G}'$ if and only if $\{(F_1(\mathcal{G}_i), \boldsymbol{z}_i)\}_{i=1}^{I}$ and $\{(F_1(\mathcal{G}'_j), \boldsymbol{z}'_j)\}_{j=1}^{J}$ are identical up to a rigid motion. That is, there exists a bijection $b : \{1, \dots, I\} \to \{1, \dots, J\}$ and a rigid transformation $g$ such that for all $i$,

$$F_1(\mathcal{G}_i) = F_1(\mathcal{G}'_{b(i)}), \quad \text{and} \quad \boldsymbol{z}_i = g \cdot \boldsymbol{z}'_{b(i)}.$$

Reindexing the units according to the bijection, we can assume: $F_1(\mathcal{G}_i) = F_1(\mathcal{G}'_i)$ for all $i$. Additionally, our construction includes the relative frame feature $g_i^\top g_j$ as an edge attribute. Since these are preserved between $\mathcal{G}$ and $\mathcal{G}'$, we have

$$g_i^\top g_j = g_i'^\top g_j' \quad \text{for all } i, j,$$

where $g_i$ denotes the local frame of $\mathcal{G}_i$, constructed using the method described in Appendix B. Specifically, each frame $g_i$ comprises an orthogonal matrix representing the orientation and a vector specifying the geometric center. By Lemma A.1, the equality $\left(F_1(\mathcal{G}_i), F_1(\mathcal{G}_j), g_i^\top g_j\right) = \left(F_1(\mathcal{G}'_i), F_1(\mathcal{G}'_j), g_i'^\top g_j'\right)$ implies that the underlying point clouds of $\mathcal{G}_i \cup \mathcal{G}_j$ and $\mathcal{G}'_i \cup \mathcal{G}'_j$ are identical up to an isometry. Using an inductive argument, we conclude that the union of all point clouds across the units $\mathcal{G}_i$ and $\mathcal{G}'_i$ must also be identical up to an isometry. Thus, the full point clouds $(\boldsymbol{X}, \boldsymbol{F})$ and $(\boldsymbol{X}', \boldsymbol{F}')$ must be identical up to a global isometry. $\square$

**Lemma A.1.** *The 3-tuple $\left(F_1(\mathcal{G}_i), F_1(\mathcal{G}_j), g_i^\top g_j\right)$ uniquely determines the union of the underlying point clouds of $\mathcal{G}_i, \mathcal{G}_j$ up to an isometry.*

*Proof.* Let $\mathcal{P}_i := \{(\boldsymbol{x}_{i,k}, \boldsymbol{f}_{i,k})\}$ and $\mathcal{P}_j := \{(\boldsymbol{x}_{j,k}, \boldsymbol{f}_{j,k})\}$ denote the underlying point clouds of $\mathcal{G}_i$ and $\mathcal{G}_j$, respectively. Let $\mathcal{F}$ be the function used to generate the equivariant local frame for each secondary structure unit, as described in Appendix B. That is, $g_i = \mathcal{F}(\mathcal{G}_i) = \mathcal{F}(\mathcal{P}_i)$ for any $i$. Our goal is to show that the union $\mathcal{P}_i \cup \mathcal{P}_j$ is uniquely determined, up to an isometry, by the 5-tuple $\left(F_1(\mathcal{G}_i), \boldsymbol{z}_i, F_1(\mathcal{G}_j), \boldsymbol{z}_j, g_i^\top g_j\right)$.

We first observe that the function $c$, which maps the point cloud $\mathcal{P}_i$ to $\mathcal{F}(\mathcal{G}_i)^\top \cdot \mathcal{P}_i := \left\{\left(\mathcal{F}(\mathcal{G}_i)^\top \cdot \boldsymbol{x}_{i,k}, \boldsymbol{f}_{i,k}\right)\right\}$, is invariant under any rigid motion applied to $\{\boldsymbol{x}_{i,k}\}$. To see this, consider a rigid motion $g \in \mathrm{E}(3)$, where $\mathrm{E}(3)$ denotes the Euclidean group of isometries. By the equivariance of $\mathcal{F}$, we have:

$$\left(\mathcal{F}(g \cdot \mathcal{G}_i)\right)^\top g \cdot \boldsymbol{x}_{i,k} = \left(\mathcal{F}(\mathcal{G}_i)\right)^\top g^\top g \cdot \boldsymbol{x}_{i,k} = \mathcal{F}(\mathcal{G}_i)^\top \cdot \boldsymbol{x}_{i,k},$$

which confirms that $c$ is invariant.

By Theorem 2.1, there exists a function $h$ such that $c(\mathcal{P}_i) = h\left(F_1(\mathcal{G}_i)\right)$ for any point cloud. Now consider the following construction that combines the information in the 3-tuple.

$$\begin{aligned}
h\left(F_1(\mathcal{G}_i)\right) \cup \left(g_i^\top g_j \cdot h\left(F_1(\mathcal{G}_j)\right)\right) &= c(\mathcal{P}_i) \cup \left(g_i^\top g_j \cdot c(\mathcal{P}_j)\right) \\
&= \left(g_i^\top \mathcal{P}_i\right) \cup \left(g_i^\top g_j \cdot g_j^\top \mathcal{P}_j\right) \quad (8) \\
&= g_i^\top \cdot \left(\mathcal{P}_i \cup \mathcal{P}_j\right).
\end{aligned}$$

This final expression shows that the union $\mathcal{P}_i \cup \mathcal{P}_j$ is uniquely determined up to a global isometry by the 3-tuple data, completing the proof. $\square$

## B Construction of Local Frames

In [9, 8], the coordinates of nodes $\boldsymbol{x}_i$ and $\boldsymbol{x}_j$ on an edge $(i, j)$ are used to define equivariant local frames, with the following transformation:

$$(\boldsymbol{x}_i, \boldsymbol{x}_j) \mapsto \left[ \frac{\boldsymbol{x}_i}{\|\boldsymbol{x}_i\|}, \frac{\boldsymbol{x}_j}{\|\boldsymbol{x}_j\|}, \frac{\boldsymbol{x}_i \times \boldsymbol{x}_j}{\|\boldsymbol{x}_i\|\|\boldsymbol{x}_j\|} \right].$$

This transformation creates a local frame for the edge, converting equivariant features within a node's neighborhood into invariant features (independent of orientation), or vice versa, through multiplication by the frame or its inverse. It can also be used to compute the crucial transition information $g_i^\top g_j$ for neighborhoods $\mathcal{N}_i$ and $\mathcal{N}_j$.

However, this method is limited to pairs of nodes and cannot be directly extended to structural units, which typically consist of more than two unordered nodes. To address this, we adopt the approach in [2], which constructs equivariant frames for point clouds (or their corresponding geometric graphs) of arbitrary size. Specifically, the algorithm in [2] constructs a frame $\mathcal{F} : \mathcal{X} \to \mathrm{E}(3)$ over the set of point clouds $\mathcal{X}$. This mapping produces not only an orthogonal matrix representing the orientation but also the geometric center of the point cloud, jointly forming a complete frame in Euclidean space. Importantly, the frame construction satisfies equivariance with respect to isometries: for any rigid motion $g \in \mathrm{E}(3)$ and any point cloud $\mathcal{P} \in \mathcal{X}$, we have

$$\mathcal{F}(g \cdot \mathcal{P}) = g \cdot \mathcal{F}(\mathcal{P}),$$

where $\cdot$ denotes the group product in $\mathrm{E}(3)$.

*Remark* B.1. While this construction applies to all point clouds, it may not be fully equivariant but rather *relaxed-equivariant* for symmetric inputs. This does not impact the distinguishability of the framework, as shown in Theorem 4.2, but it could reduce the framework's strict invariance. Fortunately, symmetric inputs are rare in practice—particularly for individual protein structures—and this issue can be mitigated by introducing small perturbations to break the symmetry.

## C SCHull Graphs

In this section, we provide a brief review of the SCHull algorithm—proposed in [41]—for constructing a sparse, connected, and rigid for a given point cloud $(\boldsymbol{X}, \boldsymbol{F})$ be a point cloud, where $\boldsymbol{X} = [\boldsymbol{x}_1, \ldots, \boldsymbol{x}_N]$, $\boldsymbol{F}$ denote the point coordinates and features, respectively. Let $\mathcal{V}$ denote the point index set. In particular, the SCHull graph for the point cloud $(\mathcal{V}, \boldsymbol{X})$ is constructing in the following three steps:

- **Step 1: Project points onto the unit sphere.** Let

$$\overline{\boldsymbol{x}} := \frac{1}{N} \sum \boldsymbol{x}_i$$

be the center of point cloud. Consider the projection

$$p_{\overline{\boldsymbol{x}}} : \mathbb{R}^3 \to \mathbb{S}^2 : \boldsymbol{x} \mapsto \frac{\boldsymbol{x} - \overline{\boldsymbol{x}}}{\|\boldsymbol{x} - \overline{\boldsymbol{x}}\|},$$

where $\mathbb{S}^2 := \{\boldsymbol{x} \in \mathbb{R}^3 \mid \|\boldsymbol{x}\| = 1\}$ is the unit sphere. That is, $p_{\overline{\boldsymbol{x}}}$ projects points onto the unit sphere centered at $\overline{\boldsymbol{x}}$. Applying this projection to all points, we obtain a new point cloud $(\mathcal{V}, p_{\overline{\boldsymbol{x}}}(\boldsymbol{X}))$ on $\mathbb{S}^2$, where

$$p_{\overline{\boldsymbol{x}}}(\boldsymbol{X}) = [p_{\overline{\boldsymbol{x}}}(\boldsymbol{x}_1), p_{\overline{\boldsymbol{x}}}(\boldsymbol{x}_2), \ldots, p_{\overline{\boldsymbol{x}}}(\boldsymbol{x}_N)].$$

- **Step 2: Construct the convex hull of the projected point cloud.** Next, SCHull constructs a convex hull—using the QuickHull algorithm [3]—for the projected point cloud $(\mathcal{V}, p_{\overline{\boldsymbol{x}}}(\boldsymbol{X}))$. Notice that this step is very efficient with a computational complexity $\mathcal{O}(N \log N)$.

- **Step 3: Construct the SCHull graph.** The SCHull graph for the given point cloud $(\boldsymbol{X}, \boldsymbol{F})$ is then defined as $\mathcal{G} = (\mathcal{V}, \mathcal{E}, \boldsymbol{F}')$. Specifically, nodes $i, j$ are connected by an edge in $E$ if and only if their projected points on the unit sphere $p_{\overline{\boldsymbol{x}}}(\boldsymbol{x}_i), p_{\overline{\boldsymbol{x}}}(\boldsymbol{x}_j)$ are connected by an

edge on the convex hull. In addition, the graph incorporates geometric attributes as follows. Each node feature in $\boldsymbol{F}'$ includes the original feature from $\boldsymbol{F}$, augmented with a scalar node attribute defined below. Similarly, each edge in $\mathcal{E}$ is associated with the following attributes:

$$\text{the edge attributes of } (i,j) : (\|\boldsymbol{x}_i - \boldsymbol{x}_j\|, \tau_{ij}) \text{ for any } (i,j) \in \mathcal{E}, \text{and}$$
$$\text{the node attributes } : \|\boldsymbol{x}_i - \overline{\boldsymbol{x}}\| \text{ for any } i \in \mathcal{V}. \tag{9}$$

SCHull graph has two remarkable properties with provable guarantees: (1) The graph is sparse and connected with edges that satisfy $|\mathcal{E}| \leq 3N - 6$ when $N = |\mathcal{V}| \geq 3$ (cf. [41, Proposition 3.1]). (2) SCHull graphs of any two non-isomorphic generic point clouds can be distinguished by a maximally expressive GNN with depth 1 (see Theorem 2.1, i.e., [41, Theorem 3.6]).

# D    DSSP Algorithm

DSSP is both a database of secondary structure assignments for all protein entries in the Protein Data Bank (PDB) [4] as well as a program that calculates DSSP entries from PDB entries [21, 34]. The algorithm first detects the presence of backbone-backbone hydrogen bonds (H-bonds). An H-bond between amino acids is considered to be present if the electrostatic interaction energy, $E$, between the carboxyl group of one and the amino group of another is calculated to be less than -0.5 kcal/mol. Specifically, $E = 0.084 \left[ \frac{1}{r(ON)} + \frac{1}{r(CH)} - \frac{1}{r(OH)} - \frac{1}{r(CN)} \right] \cdot 332$ kcal/mol where r(AB) is the interatomic distance between A and B. Following [21], Hbond($i, j$) denotes that an H-Bond is present between the carboxyl group of residue $i$ and the amino group of residue $j$.

Once H-bond presence is decided, the algorithm determines the presence of elementary H-bond patterns: $n$-turns (where $n$=3, 4, or 5) and bridges (which can be parallel or antiparallel). An $n$-turn is considered to exist at residue $i$ if Hbond($i, i+n$) is present. A bridge may exist between two non-overlapping stretches of three residues each. A parallel bridge is said to be present if Hbond($i-1, j$) and Hbond($j, i+1$) or if Hbond($j-1, i$) and Hbond($i, j+1$). An antiparallel bridge is said to be present if Hbond($i, j$) and Hbond($j, i$) or if Hbond($i-1, j+1$) and Hbond($j-1, i+1$).

These patterns are then used to identify cooperative H-bond patterns: helices, $\beta$-ladders, and $\beta$-sheets. Helices consist of two consecutive $n$-turns for fixed $n$, e.g., a 4-helix is present from residue $i$ to $i+3$ if there is a 4-turn at residue $i-1$ and another at residue $i$. Note that a 3-helix is commonly called a $3_{10}$-helix, a 4-helix an $\alpha$-helix, and a 5-helix a $\pi$-helix. $\beta$-ladders consist of one or more consecutive bridges of identical type and $\beta$-sheets consist of one or more ladders connected by shared residues. A group of five residues with high curvature is known as bend. The curvature at residue $i$ is calculated as the angle between the backbone direction of the first three and last three residues of this group of five. Specifically, if $C_j^\alpha$ is the position vector of the $\alpha$-carbon of residue $j$, a bend is considered to exist if the angle between $C_i^\alpha - C_{i-2}^\alpha$ and $C_{i+2}^\alpha - C_i^\alpha$ is greater than $70°$.

Although it is possible for an amino acid to belong to more than one of these structures, each residue is assigned a single letter from the list in Table 1. The original algorithm assigned letters in the following order from left to right: H, B, E, G, I, T, S; once a residue is assigned a letter, it is not changed. More recent versions assign $\pi$-helices before $\alpha$-helices [34] and also detect another type of helix known as a $\kappa$-helix or a poly-proline II (PPII) helix [30].

# E    Additional Experiments Details

## E.1    Datasets and Experiment Overview

### E.1.1    Datasets

**Reaction dataset**. For the reaction classification task, 3D structures of 37,428 proteins corresponding to 384 enzyme commission (EC) numbers are obtained from the Protein Data Bank, with EC annotations for each protein retrieved from the SIFTS database [7]. The dataset is divided into 29,215 proteins for training, 2,562 for validation, and 5,651 for testing. Each EC number is represented across all three splits, and protein chains sharing more than 50% sequence similarity are grouped.

**LBA dataset**. Following [18], we perform ligand binding affinity predictions on a subset of the commonly-used PDBbind refined set [39, 27]. The curated dataset of 3,507 complexes is split into

train/val/test splits based on a 30% sequence identity threshold to verify the model generalization ability for unseen proteins. For a protein-ligand complex, we predict the negative log-transformed binding affinity $pK = -\log_{10}(K)$ in molar units.

### E.1.2  Experiment Overview

We primarily follow the GNN architectures, training setups, and hyperparameter search spaces used in the baseline models GVP-GNN [18] and ProNet [37]. Our SSHG model adopts nearly identical feature embedding functions, message passing blocks, and readout functions for the hierarchical geometric graphs—namely, the Intra-Structural Graph and Inter-Structural Graph. Furthermore, we integrate Mamba [13] to capture the sequential information of the tokens in the Inter-Structural Graph. See E.2 and E.3 for details.

## E.2  Model Configuration

We integrate our SSHG framework with two models tailored for protein tasks, one is GVPNet [18] and the other is ProNet [37]. Moreover, to capture the sequential information of the secondary structure tokens, we integrate Mamba [13] into our SSHG model. See E.3.1 for further details. The illustration of the architectures of our SSHG model is shown in Figure 5. Below are some details:

- Message Passing Blocks: We use the same message-passing GNN (MPGNN) architectures as those in the baseline models [18, 37].

- Edge Feature Function: We use the same edge feature construction methods as those in the baseline models [18, 37].

- Scatter: The tensor output from the message passing blocks contains embedding vectors for all nodes across all graphs in the batch. We use the PyTorch scatter function to aggregate these node embeddings into a tensor with one embedding per graph, corresponding to the number of graphs in the batch. Similarly, we apply scatter to aggregate intra-structural graph node embeddings into a tensor with one embedding per unit in the inter-structural graph.

## E.3  Implementation Details

### E.3.1  Integrate Mamba into SSHG

To capture the sequential dependencies among secondary structure tokens in our SSHG model, we consider using Mamba. Mamba [13] is a special type of state space model defined by the following ODE system:

$$\frac{d\boldsymbol{h}(t)}{dt} = \boldsymbol{\Lambda}\Big(\boldsymbol{x}(t)\Big)\boldsymbol{h}(t) + \boldsymbol{B}\Big(\boldsymbol{x}(t)\Big)\boldsymbol{x}(t)$$
$$\boldsymbol{y}(t) = \boldsymbol{W}_o\boldsymbol{h}(t) \tag{10}$$

where:

- $t$ denotes the time step (discrete or continuous).

- $\boldsymbol{x}_t$ denotes the feature vector at time step $t$ in the input feature sequence.

- $\boldsymbol{h}(t)$ denotes the state vector, where the IC $\boldsymbol{h}(0)$ is often a learnable vector.

- $\boldsymbol{\Lambda}\big(\boldsymbol{x}(t)\big) \coloneqq \mathrm{diag}\big[\sigma\big(\boldsymbol{W}_\lambda\boldsymbol{x}(t)\big) - \mathbf{1}\big]/dt$ is the input-conditioned decay/filter coefficient, where $\boldsymbol{W}_\lambda$ is time-invariant learnable parameter and $\sigma$ denotes the sigmoid function.

- $\boldsymbol{B}\big(\boldsymbol{x}(t)\big) \coloneqq \boldsymbol{W}_B\boldsymbol{x}(t)/dt$, where $\boldsymbol{W}_B$ is a time-invariant learnable parameter.

- $\boldsymbol{W}_o$ is a time-invariant learnable parameter.

It has been increasingly adopted in large language models (LLMs) [5, 25, 35] due to its efficiency and competitive performance compared to traditional Transformer architectures.

**SSHG+Mamba**: Let $\boldsymbol{x}_i \coloneqq s_i^{(0)} = \mathrm{readout}_1(\{\!\{\boldsymbol{f}_k^{(T_1)} \mid k \in \mathcal{V}(\mathcal{G}_i)\}\!\})$ be the initial node

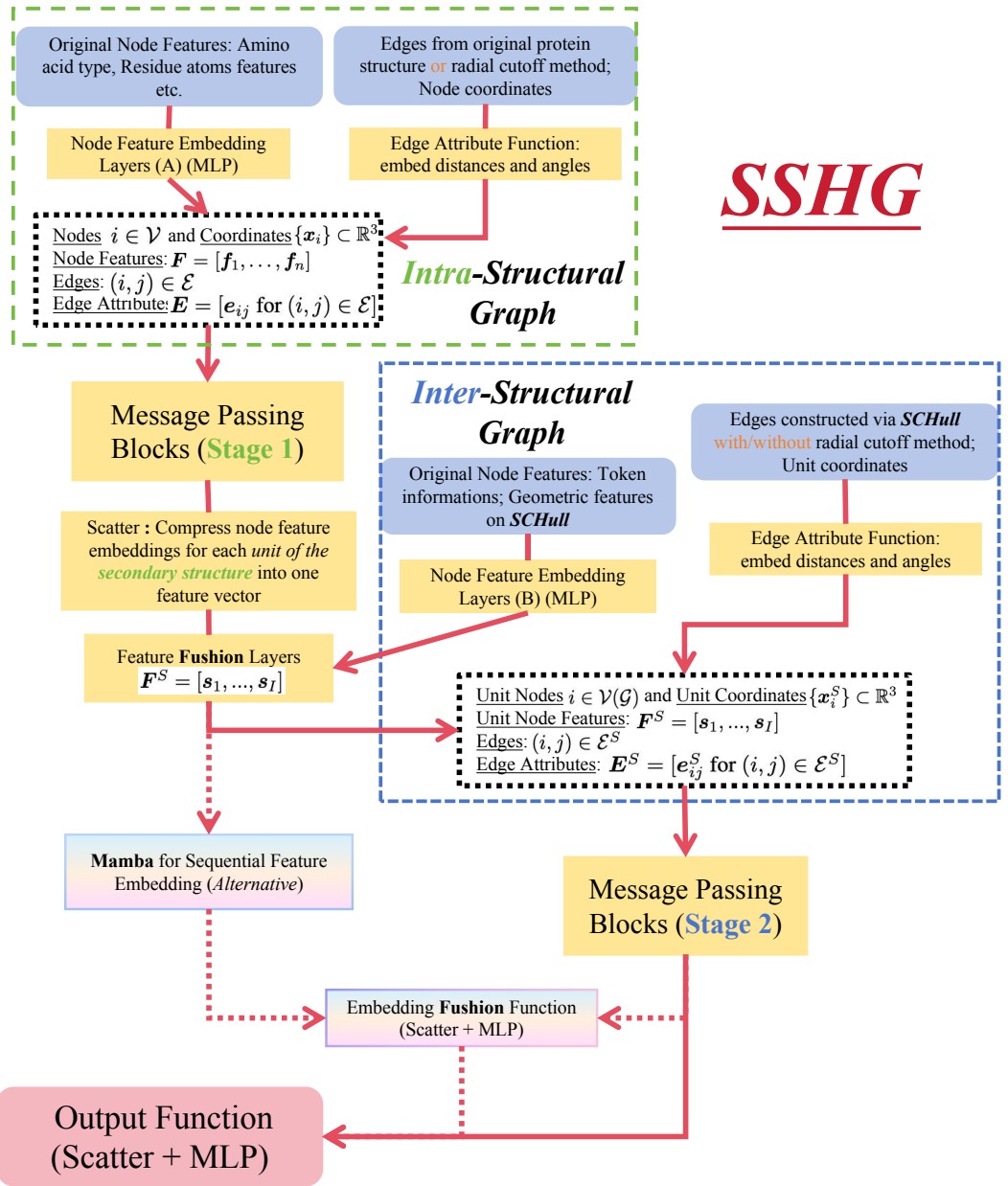

Figure 5: Illustration of the architectures of our SSHG model, with and without the integration of Mamba. The red arrows and red dashed arrows indicate the input-output dependencies in the SSHG and SSHG+Mamba models, respectively.

features for node $i$ of the inter-structural graph as defined in equation 3. Then we discretise equation 10 into

$$
\begin{aligned}
\boldsymbol{h}_{i+1} &= \bar{\boldsymbol{A}}(\boldsymbol{x}_i)\boldsymbol{h}_i + \bar{\boldsymbol{B}}(\boldsymbol{x}_i)\boldsymbol{x}_i \\
\boldsymbol{y}_{i+1} &= \bar{\boldsymbol{C}}(\boldsymbol{x}_i)\boldsymbol{h}_{i+1}
\end{aligned}
\tag{11}
$$

and then obtain the outputs $\boldsymbol{s}_i^{ssm} = \boldsymbol{y}_{i+1}$ for $i \in \mathcal{V}(\mathcal{G})$.

Back to the message passing across structural units, we obtain the node features of the inter-structural graph $\boldsymbol{s}_{global} = \text{readout}_2(\{\!\{\boldsymbol{s}_i^{(T_2)} \mid i \in \mathcal{V}(\mathcal{G})\}\!\})$ in equation 3. Then we input $\boldsymbol{s}_i^{ssm}$ and $\boldsymbol{s}_{global}$ into an output function tailored for the task's target final output. Figure 5 shows the architecture of our model integrated with Mamba.

### E.4 Discussion on Mamba

We observe that directly applying Mamba to the node sequences of the original protein graph increases the sequence length by over threefold, significantly slowing down the model. Despite this, the Mamba-only model achieves a performance comparable to GNN-based methods such as ProNet, even without leveraging geometric information. This efficiency is largely due to `mamba_ssm`, a highly optimized CUDA C++ implementation that enables fast training of complex sequence models. In contrast, most GNN implementations are primarily in Python, which introduces overhead due to slower data processing and iterative computation.

To illustrate the difference, consider typical pipeline structures:

- **GNN-based models (e.g., ProNet):**
    - Feature embedding $\rightarrow$ Python
    - Iteration over blocks $\rightarrow$ Python
    - Message passing $\rightarrow$ Python
    - Output layer $\rightarrow$ Python

- **Mamba_ssm-based models:**
    - Feature embedding $\rightarrow$ Python
    - Iteration over time steps $\rightarrow$ CUDA C++ (fused kernel)
    - Selective SSM operations $\rightarrow$ CUDA C++
    - Output layer $\rightarrow$ Python

The expensive recurrent and state-space computations in Mamba are fused into CUDA kernels, bypassing Python's loop overhead. As a result, `mamba_ssm` narrows the performance gap without relying on structural priors. While it achieves results comparable to other baselines, we still observe significant improvements when incorporating SSHG.

### E.4.1 Architecture and Experimental Setup

The number of message passing blocks, hidden channels, and dropout rates used for training SSHG on different tasks are listed in Table 6. The implementation of our methods is based on PyTorch and Pytorch Geometric, and all models are trained with the Adam optimizer. All are conducted on a single NVIDIA GeForce RTX 3090 24 GB. The hyperparameter searching space for training is shown in Table 7.

| Hyperparameter | Values/Search Space | |
| --- | --- | --- |
| | **React** | **LBA** |
| Number of layers (1st) | 1, 2 | 1, 2 |
| Number of layers (2nd) | 2, 3 | 2, 3 |
| Hidden channels | 64, 128, 256 | 128, 192, 256 |
| Dropout | 0.2, 0.3, 0.5 | 0.2, 0.3 |
| Mamba Blocks | 4 | 4 |

Table 6: Model hyperparameters for SSHG

| Hyperparameter | Values/Search Space | |
| --- | --- | --- |
| | **React** | **LBA** |
| Epochs | 500, 1000 | 300, 500 |
| Batch size | 16, 32 | 8, 16, 32 |
| Learning rate | 1e-4, 5e-4 | 5e-5, 1e-4, 2e-4 |
| Learning rate scheduler | steplr | steplr |
| Learning rate decay factor | 0.5 | 0.5 |
| Learning rate decay epochs | 50, 100 | 50, 100 |

Table 7: Training hyperparameters search space.

## E.5 Additional Ablation Studies

| | w/ SS | w/ hierarchical | w/ geometry | Test Acc |
|---|---|---|---|---|
| ProNet | ✗ | ✗ | ✗ | 86.4 |
| – | ✓ | ✗ | ✗ | 87.0 |
| – | – | ✓ | ✗ | 87.2 |
| – | – | ✓ | ✓ | 87.5 |
| GVPGNN[18] | ✗ | ✗ | ✗ | 68.5 |
| – | ✓ | ✗ | ✗ | 66.7 |
| – | – | ✓ | ✗ | 71.5 |
| – | – | ✓ | ✓ | 73.6 |

Table 8: **Feature Selection**: Different GNNs with (w/) or without (w/o) hierarchical geometric graphs, geometric features $g_i^\top g_j$, or secondary structure tokens.

Table 8 shows that simply appending SS tokens as a feature to the original GNNs does not necessarily improve performance. In contrast, combining SS information and geometric features through a dedicated hierarchical mechanism leads to consistent improvements.

