# OpenReview forum: "Towards Multiscale Graph-based Protein Learning with Geometric Secondary Structural Motifs"
_NeurIPS.cc/2025/Conference — NeurIPS 2025 poster_

### Official Review · Reviewer_kpTM · 2025-06-29

**Clarity:** 3
**Significance:** 3
**Originality:** 2
**Rating:** 5
**Confidence:** 4

**Summary:**

The authors propose a hierarchical method for modeling proteins by grouping residues into contiguous units that share the same secondary structure. They develop an architecture which first models each secondary structure unit with a GNN, then models the relationship between units with another GNN. The method is applied to two problems using two different GNN architectures and shows improvements over the baseline in two benchmark tasks.

**Questions:**

1. Fix the set of models included in the benchmarks (GVP-GNN w/ SSHG, Mamba-backbone).
2. Include more results or analysis in the paper. In particular, capability improvements on multiple would improve the significance score, new capabilities that were out of reach due to memory issues would improve the significance score, and evaluating different ways creating hierarchical graphs on proteins would improve the originality score. Other options also exist, if the authors have ideas, I again don't want to be too prescriptive.

**Ethical Concerns:**

["NO or VERY MINOR ethics concerns only"]

**Final Justification:**

See comment on author submission.

**Limitations:**

yes

**Quality:**

3

**Strengths And Weaknesses:**

## Strengths
The paper presents a simple, widely applicable idea (break a problem into hierarchical graphs where the breaks are defined by prior domain knowledge).

It then demonstrates the performance improvements not only on different tasks, but across multiple backbone architectures (caveat: the performance of Mamba-backbone is not shown, I think it needs to be).

## Weaknesses
First - there appear to be some missing comparisons in the experiments. The authors say they applied SSHG to the GVP-GNN, but no results are shown in the table (these seem to appear only in the appendix ablations, which is a bit confusing). They authors also show Mamba-SSHG but not Mamba-backbone, so it is not possible to say whether Mamba-backbone would be more performant or less.

Second - the paper overall seems just a bit light on content. It's a pretty simple idea, with a pretty straightforward implementation (which I'm not opposed to! I like simple ideas that work well). Still, I'd expect this to be made up for by more of something - more benchmarks, more ablations, more analysis. As a comparison, one paper the authors build on, ProNet, has 4 benchmark tasks with multiple splits (in comparison to the 2 from this paper).

I don't want to be hugely prescriptive of what the authors could do, but here are a few ideas:

* Include more benchmarks, at least get parity with what similar papers report
* Ablate the graph construction: what happens if you use 3-class secondary structure instead of 8-class? What happens if you assign classes based on foldseek structure discretization? Is it possible to construct the graph based on TERMs (from the Grigoriyan Lab: https://grigoryanlab.org/terms/)
* Analyze what is learned by the model that *isn't* learned by backbone models. Are there particular structures it performs really well on? Do you observe increased generalization with lower sequence identity cutoffs?
* Show capabilities on proteins that weren't previously possible. Does the decreased memory enable processing very large protein structures?

---

> ### Author Rebuttal · Authors · 2025-07-31
>
> We sincerely appreciate the reviewer’s acknowledgment, insightful comments, and detailed suggestions for improving our paper.
> We have carefully followed your feedback and addressed your concerns as outlined below:
>
> ---
> **Q1. First - there appear to be some missing comparisons in the experiments. The authors say they applied SSHG to the GVP-GNN, but no results are shown in the table (these seem to appear only in the appendix ablations, which is a bit confusing). They authors also show Mamba-SSHG but not Mamba-backbone, so it is not possible to say whether Mamba-backbone would be more performant or less.**
>
> **Response:**
>
> Thank you for highlighting this important point. We have now included a complete set of results for both GVP-GNN and Mamba backbones, with and without SSHG, in the revised paper. This ensures a more comprehensive and transparent comparison. Importantly, we emphasize the impact of integrating SSHG into the GVP-GNN model for EC classification. SSHG reduces the average number of edges from 14,881 to 1,593—leading to  a **~87% reduction in memory usage** (from 11,248 MiB to 1,416 MiB) and a **33% reduction in training time** per epoch (from 354s to 236s). At the same time, test accuracy improves from 69.2% to 73.6%. This demonstrates that SSHG not only brings significant efficiency gains  but also  improves accuracy.
>
> **LBA**
> | Model   | RMSE ↓  | Pearson ↑ | Spearman ↑    | Ave.Time  ↓    |
> |--|----|--|-|---|
> | GVP-GNN                | 1.529  | 0.441| 0.432  |   49
> | GVP-GNN + SSHG     | **1.488**  | **0.512**   | **0.477** | **35**
> | | | |
> | Mamba  | 1.457 ± 0.004 | 0.565 ±  0.003    | 0.554 ±  0.004    | **27**
> | Mamba + SSHG  | **1.399 ± 0.003**        | **0.614 ±  0.003**    | **0.610 ±  0.003**    | 29
>
> **EC**
> | Model  | Acc |  Ave.Time   |
> | -|- | -|
> |  Mamba    |85.9  |  236    |
> |  Mamba + SSHG   |  88.4| 157    |
>
> **Extended Ablation Study on EC**
> | Model   | +SSHG   | Cutoff | Avg. Num Edges | Time (s/epoch) ↓ | Mem (MiB) ↓ | Test Acc (%) ↑ | Acc/Time (×100) ↑ |
> | --| -| -| - | -- | -- | - | - |
> | GVP-GNN | No      | 4      | 1,034.5  | 216 | 1,558       | 65.5  | 30.3  |
> |  | No      | 6      | 4,755.2   | 254  | 3,828       | 66.9  | 26.3   |
> |  | No      | 8      | 8,013.9   | 298  | 6,286       | 68.1  | 22.9  |
> |  | No      | 10     | 11,316.8    | 334  | 8,930       | 68.5   | 20.5 |
> |  | No      | 16     | 14,881.1  | 354 | 11,248      | 69.2 | 19.5  |
> |   | **Yes** | —      | 1,593.3  | 236   | 1,416       | **73.6** | **31.2**    |
>
> We appreciate the reviewer’s attention to detail and will ensure that all tables are updated in the revised manuscript to include complete comparisons across all model variants.
>
> ---
> **Q2. Second - the paper overall seems just a bit light on content. It's a pretty simple idea, with a pretty straightforward implementation (which I'm not opposed to! I like simple ideas that work well). Still, I'd expect this to be made up for by more of something - more benchmarks, more ablations, more analysis. As a comparison, one paper the authors build on, ProNet, has 4 benchmark tasks with multiple splits (in comparison to the 2 from this paper).**
>
> **Response:** We thank the reviewer for the detailed and constructive feedback. We greatly appreciate the recognition of the simplicity and potential of our approach, and we agree that further content and analysis would strengthen the manuscript. Below, we summarize our additional experiments:
>
> **Ablation Studies:**
> We have included comparisons with Mamba-backbone and GVP-GNN w/ SSHG to clarify the performance improvements gained through SSHG as mentioned above. Moreover, we have expanded our experiments by including more ablation studies for ProNet and GVP-GNN models on the EC benchmark. These results demonstrate that integrating SSHG consistently improves both accuracy and efficiency. For example, on the ProNet model, integrating SSHG drastically reduces the number of edges from 14,881 to 1,593—yielding nearly a **2× speedup in training time** per epoch (247s down to 140s) and a **90% reduction in memory usage** (17,768 MiB down to 1,818 MiB). This is achieved while improving accuracy slightly from 87.0% to 87.2%. Such results represent a substantial and meaningful breakthrough in balancing efficiency and accuracy, clearly demonstrating the practical value and impact of SSHG.
>
> **Extended Ablation Study on EC**
> | Model   | +SSHG   | Cutoff | Avg. Num Edges | Time (s/epoch) ↓ | Mem (MiB) ↓ | Test Acc (%) ↑ | Acc/Time (×100) ↑ |
> | ------- | ------- | ------ | -------------- | ---------------- | ----------- | -------------- | ----------------- |
> | ProNet  | No      | 4      | 1,034.5        | 138              | 1,290       | 78.1           | 56.6              |
> |         | No      | 6      | 4,755.2        | 165              | 7,760       | 82.1           | 49.8              |
> |         | No      | 8      | 8,013.9        | 185              | 9,580       | 85.6           | 46.3              |
> |         | No      | 10     | 11,316.8       | 210              | 14,548      | 86.4           | 41.1              |
> |         | No      | 16     | 14,881.1       | 247              | 17,768      | 87.0           | 35.2              |
> |         | **Yes** | —      | 1,593.3        | 140              | 1,818       | **87.2**       | **62.3**          |
>
> **Other Baselines:** to address additional reviewers' concerns,
> we have now included a comparison between GCN models with and without the SSHG framework on both the EC and LBA benchmarks. We observe consistent improvements in terms of both accuracy and average runtime, demonstrating the general applicability and efficiency of our method.
>
> **EC**
> |   | Test Acc | Ave.Time |
> |-|-|-|
> | GCN|68.5|186|
> | GCN + SSHG|**71.2**| **150**|
>
> **LBA**
> | |  RMSE ↓|Pearson  ↑ | Spearman  ↑ | Ave.Time |
> |-|--|-|-|-|
> | GCN |  1.925 | 0.322  | 0.287 | 28|
> | GCN + SSHG |**1.788** | **0.392**|**0.359**| **23** |
>
> We are currently training SSHG-enhanced models for the Protein–Protein Interaction (PPI) task in ATOM3D. As these models are being trained from scratch and require extensive time and tuning, obtaining high-quality results within a week is challenging. We plan to include PPI evaluation in future versions and will mention this ongoing extension in the revised manuscript’s conclusion. In addition, we agree that studying the impact of different motif definitions is valuable. We are particularly interested in the reviewer’s suggestions of using 3-class secondary structure. However, implementing and validating these alternatives would require significant additional work, especially around preprocessing pipelines, so we plan to explore them as part of our future work. We will explicitly state this in the discussion section.
>
>
> We appreciate the reviewer’s thoughtful suggestions and will continue building on them to improve both the clarity and depth of the work.

---

> > ### Comment · Reviewer_kpTM · 2025-08-05
> > **Thank you for your response and updates**
> >
> > Thanks for the detailed updates and benchmarking. On balance I am leaning towards accepting this paper, and I'll change my score to accept.
> >
> > It is a simple idea, but it is clearly widely applicable across many architectures and seems surprisingly impactful.
> >
> > For the authors, I think this paper / ideas could be significantly enhanced with more creativity in what is explored and presented beyond headline benchmark numbers. Dig into what is learned vs. a more network, what kind of bias this introduces and why that's important, etc. I wouldn't be surprised if there was an even better way to group residues than 8 class secondary structure - perhaps even a learned grouping.
> >
> > Regardless, I think the idea is worth sharing, so I'll set my score to accept.

---

> > > ### Author Response · Authors · 2025-08-05
> > > **Thank you**
> > >
> > > We’re grateful for your attention to our rebuttal and your continued support. We also deeply appreciate your valuable suggestions.

---

### Official Review · Reviewer_Ckw3 · 2025-07-01

**Clarity:** 3
**Significance:** 3
**Originality:** 3
**Rating:** 4
**Confidence:** 3

**Summary:**

This paper proposes a framework for graph-based protein learning, SSHG (Secondary Structure-based Hierarchical Graph). It bases protein secondary structures (α-helices, β-strands) to build hierarchical graph representations. The method constructs two-level graphs: fine-grained subgraphs within each secondary motif and a coarse-grained graph connecting motifs. A two-stage GNN processes these hierarchical graphs to capture both local and global features. Experiments on enzyme classification and ligand binding affinity show improved accuracy and reduced training time compared to baselines.

**Questions:**

What specific GNN architecture is used in Mamba+SSHG, and how are the sequential dependencies handled?
Could you add more evaluation tasks, like Protein-Protein Interface (PPI) in ATOM3D? This task is also used by ProNet and GVP-GNN.
How does your method compare to simpler long-range modeling methods like adding virtual nodes or fully-adjacent layers?

**Ethical Concerns:**

["NO or VERY MINOR ethics concerns only"]

**Final Justification:**

The authors have answered my most significant requests for clarifications and I also appreciate their efforts of answering the other reviews. I think the paper has gained in clarity and have updated my score accordingly

**Limitations:**

yes

**Quality:**

3

**Strengths And Weaknesses:**

Strengths:
Clear biological motivation: Using secondary structures as hierarchical units is intuitive and biologically meaningful
Computational efficiency: Reduces edge complexity while maintaining performance
Modular design: Framework can integrate with different backbone GNNs, like GVP-GNN and ProNet
Weaknesses:
Overly complex theoretical analysis: Sections 2-3 contain extensive expressiveness theory that makes the paper hard to follow, while the core idea is simple and intuitive
Questionable theoretical assumptions: Assumption 4.1 is unrealistic with standard GNN components like commonly used pooling operations; for example, GVP-GNN uses mean pooling, which is not injective
Limited baselines: Missing comparisons with basic GNNs, like GCN or GAT. Recent work shows that these simple, basic GNNs can work quite well.
Incomplete experimental tables: Table 2 and Table 3 compared ProNet-Backbone and ProNet+ SSHG. However, only GVP-GNN and Mamba+SSHG are listed in the tables, and the corresponding GVP-GNN+SSHG and Mamba are missing for comparison.

Minor issues:
Missing equation number at line 119
Appendix C skips step 3
In the checklist, they claim that "in the conclusion section, we have listed a few potential future works", but in fact, there are none.

---

> ### Author Rebuttal · Authors · 2025-07-31
>
> We appreciate the reviewer’s insightful comments. Below, we address each of your concerns one by one:
>
> ---
> **Q1. Weaknesses: Overly complex theoretical analysis: Sections 2-3 contain extensive expressiveness theory that makes the paper hard to follow, while the core idea is simple and intuitive.**
>
> **Response:**
>
> We thank the reviewer for this valuable feedback. We agree that the core idea—leveraging secondary structure motifs in a multiscale GNN—is intuitive and biologically motivated. Our goal in Sections 2–3 was to provide a rigorous theoretical foundation for our hierarchical framework, particularly to support our claims of expressiveness and geometric fidelity—properties that are critical for graph learning but have not been addressed in existing motif-based approaches. Such theoretical guarantees are typically provided only in non-motif-based frameworks [16, 25, 37, 39] (as cited in our paper).
>
> To improve clarity and accessibility, we will revise the manuscript to:
> - Add a high-level intuition and summary at the beginning of Section 2 to explain the motivation and practical implications of the theory in plain language.
> - Streamline the theoretical exposition by moving detailed proofs and formal definitions (e.g., SCHull graph properties, local frame transformations) to the appendix
> - Clarify the connection between the theoretical results and the empirical benefits observed in our experiments (e.g., improved accuracy and efficiency).
>
> We believe this balance will preserve the rigor for interested readers while making the main narrative more approachable.
>
> **Q2. Questionable theoretical assumptions: Assumption 4.1 is unrealistic with standard GNN components like commonly used pooling operations; for example, GVP-GNN uses mean pooling, which is not injective.**
>
> **Response:**
>
> We appreciate the reviewer’s thoughtful observation. Indeed, Assumption 4.1—requiring injective message and aggregation functions—does not strictly hold in many practical GNN architectures, such as GVP-GNN, which employs mean pooling. We would like to clarify the motivation and scope of this assumption in our theoretical analysis.
>
> **Theoretical Motivation:**
> As discussed in the Background section, Assumption 4.1 is adopted to align with the expressive power analysis based on the Weisfeiler–Lehman (WL) test, a widely used framework for evaluating the representational capacity of GNNs [16, 25, 37, 39]. Our goal with this assumption is to characterize the best possible representational power an architecture can achieve, but not to impose a constraint on practical implementations. Similar assumptions are standard in the literature for theoretical completeness.
>
> **Practical Implementation:**
> In practice, like many prior works, we do not strictly enforce injectivity. Instead, we rely on sufficiently expressive multilayer perceptrons (MLPs) with ReLU activations and adequate width to approximate desirable mappings, as supported by prior studies. Our models—including those using non-injective pooling like mean pooling—still benefit from the SSHG framework and achieve consistent performance gains, demonstrating its practical effectiveness.
>
> **Future Directions:**
> We agree that incorporating more expressive or injective aggregation schemes (e.g., PNA) could further enhance performance. We will mention this as a promising direction for future work in the revised paper.
>
> To enhance clarity, we have revised the discussion of Assumption 4.1 in the updated manuscript.
>
> **Q3. Limited baselines: Missing comparisons with basic GNNs, like GCN or GAT. Recent work shows that these simple, basic GNNs can work quite well.**
>
> **Response:** Thank you for the helpful comment. To address your concerns, we have now included a comparison between GCN models with and without the SSHG framework on both the EC and LBA benchmarks. We observe consistent improvements in terms of both accuracy and average runtime, demonstrating the general applicability and efficiency of our method.
>
> **EC**
> |   | Test Acc | Ave.Time |
> |-|-|-|
> | GCN|68.5|186|
> | GCN + SSHG|**71.2**| **150**|
>
> **LBA**
> | |  RMSE ↓|Pearson  ↑ | Spearman  ↑ | Ave.Time |
> |-|--|-|-|-|
> | GCN |  1.925 | 0.322  | 0.287 | 28|
> | GCN + SSHG |**1.788** | **0.392**|**0.359**| **23** |
>
> ---
>
> **Q4. Incomplete experimental tables: Table 2 and Table 3 compared ProNet-Backbone and ProNet+ SSHG. However, only GVP-GNN and Mamba+SSHG are listed in the tables, and the corresponding GVP-GNN+SSHG and Mamba are missing for comparison.**
>
> **Response:**
>
> We agree that including both the original and SSHG-enhanced versions of each backbone model is essential for a complete and fair comparison. We have now complete the table for the comprehsen comparison as below. Importantly, on the GVP-GNN model for EC classification, integrating SSHG reduces the average number of edges from 14,881 to 1,593—leading to  a **~87% reduction in memory usage** (from 11,248 MiB to 1,416 MiB) and a **33% reduction in training time** per epoch (from 354s to 236s). At the same time, test accuracy improves from 69.2% to 73.6%. This demonstrates that SSHG not only brings significant efficiency gains  but also  improves accuracy.
>
> **LBA**
> | Model   | RMSE ↓  | Pearson ↑ | Spearman ↑    | Ave.Time  ↓    |
> |--|----|--|-|---|
> | GVP-GNN                | 1.529  | 0.441| 0.432  |   49
> | GVP-GNN + SSHG     | **1.488**  | **0.512**   | **0.477** | **35**
> | | | |
> | Mamba  | 1.457 ± 0.004 | 0.565 ±  0.003    | 0.554 ±  0.004    | **27**
> | Mamba + SSHG  | **1.399 ± 0.003**        | **0.614 ±  0.003**    | **0.610 ±  0.003**    | 29
>
> **EC**
> | Model  | Acc |  Ave.Time   |
> | -|- | -|
> |  Mamba |85.9  |  236    |
> |  Mamba + SSHG   |  **88.4**| **157** |
>
> **Extended Ablation Study on EC**
> | Model   | +SSHG   | Cutoff | Avg. Num Edges | Time (s/epoch) ↓ | Mem (MiB) ↓ | Test Acc (%) ↑ | Acc/Time (×100) ↑ |
> | --| -| -| - | - | -- | - | - |
> | GVP-GNN | No | 4      | 1,034.5  | 216 | 1,558 | 65.5  | 30.3  |
> |  | No | 6 | 4,755.2   | 254  | 3,828       | 66.9  | 26.3   |
> |  | No | 8| 8,013.9   | 298  | 6,286       | 68.1  | 22.9  |
> |  | No | 10| 11,316.8    | 334  | 8,930       | 68.5   | 20.5 |
> |  | No | 16 | 14,881.1  | 354 | 11,248      | 69.2 | 19.5  |
> |   | **Yes** | —      | 1,593.3  | 236   | 1,416       | **73.6** | **31.2**    |
>
> We appreciate the reviewer’s attention to detail and will ensure that all tables are updated in the revised manuscript to include complete comparisons across all model variants.
>
> **Q5. Minor issues: Missing equation number at line 119 Appendix C skips step 3 In the checklist, they claim that "in the conclusion section, we have listed a few potential future works", but in fact, there are none.**
>
> **Response:**
>
> We will address each point as follows:
> - Missing equation number at line 119: We acknowledge that the message-passing equation in Section 2 was not numbered. We will assign it a proper equation number (e.g., Eq. (1)) in the revised manuscript for clarity and consistency.
>
> - Appendix C skips step 3: Thank you for catching this. The numbering in Appendix C inadvertently jumps from Step 2 to Step 4. We will correct this to include Step 3 (which was implicitly described) and renumber the steps appropriately.
>
> - Checklist inconsistency regarding future work: You're right—the conclusion currently lacks a discussion of future work despite the checklist claim. We will revise it to briefly outline directions such as adopting injective aggregation, alternative motif definitions, and integration with protein language models.
>
> We appreciate the reviewer’s attention to detail and will ensure these corrections are made in the final version.
>
> **Q6. What specific GNN architecture is used in Mamba+SSHG, and how are the sequential dependencies handled? Could you add more evaluation tasks, like Protein-Protein Interface (PPI) in ATOM3D? This task is also used by ProNet and GVP-GNN. How does your method compare to simpler long-range modeling methods like adding virtual nodes or fully-adjacent layers?**
>
> **Response:**
>
> We observe that directly applying Mamba to the node sequences of the original protein graph increases the sequence length by over threefold, significantly slowing down the model. Despite this, the Mamba-only model achieves performance comparable to GNN-based methods such as ProNet, even without leveraging geometric information. This surprising efficiency is largely due to the use of `mamba_ssm`, a highly optimized CUDA C++ implementation that enables fast training of complex sequence models. In contrast, most GNN implementations are primarily in Python, which introduces overhead due to slower data processing and iterative computation.
>
> To illustrate the difference, consider the following typical pipeline structures:
>
> **GNN-based models (e.g., ProNet):**
>
> * Feature embedding → Python
> * Iteration over blocks → Python
> * Message passing → Python
> * Output layer → Python
>
> **Mamba\_ssm-based models:**
>
> * Feature embedding → Python
> * Iteration over time steps → CUDA C++ (fused kernel)
> * Selective SSM operations → CUDA C++
> * Output layer → Python
>
> The expensive recurrent/state-space computations in Mamba are fused into CUDA kernels, bypassing Python’s loop overhead. As a result, Mamba\_ssm narrows the performance gap without relying on structural priors. While it achieves results comparable to other baselines, we still observe significant improvements when incorporating SSHG. We are currently working on the PPI task and comparing our approach with virtual node and fully connected layers. Training all models from scratch makes it difficult to obtain meaningful results within a week. For fully connected layers, we suspect that any accuracy gains come at the significant cost of efficiency. As shown in the table above, increasing the cutoff to 16 in GVPNN significantly increases memory usage (nearly 10×) without outperforming our model. We plan to include these results in future work and will mention this ongoing extension in the revised manuscript.

---

> > ### Comment · Reviewer_Ckw3 · 2025-08-03
> >
> > Thank you for your detailed response and the new set of experiments. I think this has greatly improved your paper and the comparison to other methods is now much clearer.

---

> > > ### Author Response · Authors · 2025-08-03
> > > **Thank you and follow-up**
> > >
> > > Thank you for reviewing our rebuttal and acknowledging the improvements in our work. We note, however, that our paper remains in the borderline reject category. We would greatly appreciate your feedback on any specific concerns or questions that require further clarification.

---

> > > > ### Author Response · Authors · 2025-08-08
> > > > **Further Clarification**
> > > >
> > > > Dear Reviewer Ckw3,
> > > >
> > > > Thank you again for your thoughtful feedback and for acknowledging that our response addressed your concerns. We would be grateful if you could consider updating your score to reflect your positive assessment.
> > > >
> > > >
> > > > Regards,
> > > >
> > > > Authors

---

### Official Review · Reviewer_obWc · 2025-07-02

**Clarity:** 3
**Significance:** 2
**Originality:** 2
**Rating:** 3
**Confidence:** 4

**Summary:**

The paper proposes a multi-scale protein learning strategy via GNNs. They apply a two-level strategy to learn protein embeddings. The first is to split the proteins into a set of motifs (where each one contains rich domain-specific knowledge), and encode each motif individually. Then, they connects the motifs to form an additional graph and use another GNN to learn the final protein embedding.

**Questions:**

1. Can the authors clearly demonstrate the novelty of the paper?
2. Can the authors provide additional ablation studies to demonstrate the superiority of the method?

**Ethical Concerns:**

["NO or VERY MINOR ethics concerns only"]

**Final Justification:**

The authors provide a simple yet effective method for pretein learning using secondary motifs. Despite the authors provide some of evidences to show the superiority of the method, I still have concerns on the novelty. I think the paper is still below the bar of NeurIPS.

**Limitations:**

See weaknesses.

**Quality:**

2

**Strengths And Weaknesses:**

Strengths:
1. The paper is easy to follow.
2. The idea is straightforward and make sense.
3. The proposed method achieves good performance.

Weaknesses:
1. The work seems to be incremental. The idea of using motifs to represent a protein/molecule is widely applied in many works [1,2,3]. The authors use a widely applied technique to split a protein sequence into a set of groups, where each one represents a motif, and treat each group as a fully connected subgraph. The novelty only lies on the side of how to encode each motif.

[1] Representation for Discovery of Protein Motifs, ISMB 1993

[2] Molecular Representation Learning via Heterogeneous Motif Graph Neural Networks, ICML 2022.

[3] Molformer: Motif-based transformer on 3d heterogeneous molecular graphs, AAAI 2023.

2. The experiments are not sufficient. The authors did experiments over two benchmark datasets and also analyze the model efficiency and architecture. However, there is no clear understanding/demonstration on whether learning the motifs can facilitate the model performance and what are the trade-offs of the method.

---

> ### Author Rebuttal · Authors · 2025-07-30
>
> We appreciate the reviewer’s insightful comments on our work. We would like to clarify the key novelties of our framework and highlight the substantial improvements in both efficiency and performance that our method achieves compared to prior approaches.
>
> ----
>
> **Q1. The work seems to be incremental. The idea of using motifs to represent a protein/molecule is widely applied in many works [1,2,3]. The authors use a widely applied technique to split a protein sequence into a set of groups, where each one represents a motif, and treat each group as a fully connected subgraph. The novelty only lies on the side of how to encode each motif. Can the authors clearly demonstrate the novelty of the paper?**
>
> [1] Representation for Discovery of Protein Motifs, ISMB 1993
>
> [2] Molecular Representation Learning via Heterogeneous Motif Graph Neural Networks, ICML 2022.
>
> [3] Molformer: Motif-based transformer on 3d heterogeneous molecular graphs, AAAI 2023.
>
> **Response:**
>
> We thank the reviewer for pointing out these important and relevant motif-based works. While our framework does share the high-level idea of leveraging motifs, we respectfully clarify that our contribution goes beyond this general idea. Our key novelty lies in the **biologically grounded, hierarchical, and geometrically expressive graph construction**, as well as the theoretically supported and **scalable GNN design**. Specifically:
>
> **1. Biologically Grounded Hierarchical Graph Construction**:
> Prior works (e.g., [2][3]) represent molecules using low-level semantic units such as bonds, rings, or functional groups (e.g., carboxyl groups) as motifs. In contrast, our framework adopts biologically meaningful secondary structure elements (e.g., α-helices, β-strands), which better reflect the hierarchical structures of proteins, similar to [1].
> Unlike [1], which encodes only the sequential order of motifs, we build a two-level hierarchical graph capturing both local geometry and long-range 3D dependencies:
> - Each secondary structure motif (e.g., α-helix, β-strand) is modeled as a rigid intra-structural graph using SCHull, rather than a fully connected graph, preserving fine-grained geometric details while maintaining sparsity.
> - A coarse-grained inter-structural graph is constructed over motifs, incorporating both geometric centers and relative 3D orientations (via local frames), to capture geometric relationships and long-range dependencies among structural units.
>
> To the best of our knowledge, this is the first framework to incorporate secondary structures into motif-level graph construction for downstream GNN models.
>
> **2. Theoretical Guarantees of Expressiveness:**
> We provide a **formal expressiveness guarantee** (Theorem 4.2) showing that our two-stage GNN can **distinguish non-isomorphic protein structures up to rigid motions**—a key property of the graph learning framework not addressed in [1], [2], or [3]. Such guarantees are typically offered only in non-motif-based frameworks [16, 25, 37, 39] (cited in our paper).
>
> **3. Sparse and Scalable Design with Provable Bounds:**
> We provide a provable sparsity bound (Proposition 3.2) on the total number of edges in the hierarchical graph, which is critical for scaling to large proteins. This contrasts with prior motif-based models, which often lack such guarantees due to their design or depend on dense or heuristic graph constructions.
>
> **4. General-Purpose Framework with Empirical Gains:**
> Our SSHG framework is modular and model-agnostic, and we demonstrate consistent improvements in both accuracy and efficiency across diverse tasks (enzyme classification, ligand binding affinity) and backbones (ProNet, Mamba, GVP-GNN). This generality and empirical robustness go beyond the scope of prior motif-based models.
>
> In summary, this is the **first work** to combine:
> * Secondary structure-defined motifs,
> * Hierarchical geometric graphs with orientation-aware edges,
> * Expressiveness guarantees via WL analysis, and
> * Scalable, sparse modeling for large protein structures.
> These provide **interpretability**, **biological alignment**, and **geometric expressiveness** not seen in prior motif-based GNNs. We will revise the manuscript to more clearly articulate these distinctions and cite \[1–3] in our related work section.
> ---
>
> **Q2. The experiments are not sufficient. The authors did experiments over two benchmark datasets and also analyze the model efficiency and architecture. However, there is no clear understanding/demonstration on whether learning the motifs can facilitate the model performance and what are the trade-offs of the method. Can the authors provide additional ablation studies to demonstrate the superiority of the method?**
>
> **Response:**
>
> Our experimental design rigorously evaluates both predictive performance and computational efficiency of the proposed SSHG framework against existing methods. In response to the reviewers’ requests, we have conducted more comprehensive ablation studies, some of which are summarized below.
>
> Importantly, on the ProNet model for EC classification, integrating SSHG drastically reduces the number of edges from 14,881 to 1,593—yielding nearly a **2× speedup in training time** per epoch (247s down to 140s) and a **90% reduction in memory usage** (17,768 MiB down to 1,818 MiB). This is achieved while improving accuracy slightly from 87.0% to 87.2%. Such results represent a substantial and meaningful breakthrough in balancing efficiency and accuracy, clearly demonstrating the practical value and impact of SSHG.
>
> **Ligand Binding Affinity (LBA)**
>
> | Model                  | RMSE           ↓     | Pearson        ↑     | Spearman       ↑     |     Ave.Time  ↓    |
> |------------------------|----------------------|----------------------|----------------------|---------------|
> | GVP-GNN                | 1.529                | 0.441                | 0.432                |   49
> | GVP-GNN + SSHG     | **1.488**                | **0.512**                | **0.477**                | **35**
> | | | |
> | Mamba                  | 1.457 ± 0.004        | 0.565 ±  0.003    | 0.554 ±  0.004    | **27**
> | Mamba + SSHG           | **1.399 ± 0.003**        | **0.614 ±  0.003**    | **0.610 ±  0.003**    | 29
> | | | |
> | ProNet-Backbone        | 1.458                | 0.546                | 0.550                | 32.1
> | ProNet-Backbone + SCHull| **1.321**        | **0.581 ± 0.001**         | 0.578    | 34.4
> | ProNet + SSHG| 1.435 ± 0.004    | **0.579 ± 0.004**    | **0.591 ± 0.003**    | **24**
>
> **Extended Ablation Study on EC**
> | Model   | +SSHG   | Cutoff | Avg. Num Edges | Time (s/epoch) ↓ | Mem (MiB) ↓ | Test Acc (%) ↑ | Acc/Time (×100) ↑ |
> | ------- | ------- | ------ | -------------- | ---------------- | ----------- | -------------- | ----------------- |
> | ProNet  | No      | 4      | 1,034.5        | 138              | 1,290       | 78.1           | 56.6              |
> |         | No      | 6      | 4,755.2        | 165              | 7,760       | 82.1           | 49.8              |
> |         | No      | 8      | 8,013.9        | 185              | 9,580       | 85.6           | 46.3              |
> |         | No      | 10     | 11,316.8       | 210              | 14,548      | 86.4           | 41.1              |
> |         | No      | 16     | 14,881.1       | 247              | 17,768      | 87.0           | 35.2              |
> |         | **Yes** | —      | 1,593.3        | 140              | 1,818       | **87.2**       | **62.3**          |
> |         |         |        |                |                  |             |                |                   |
> | GVP-GNN | No      | 4      | 1,034.5        | 216              | 1,558       | 65.5           | 30.3              |
> |         | No      | 6      | 4,755.2        | 254              | 3,828       | 66.9           | 26.3              |
> |         | No      | 8      | 8,013.9        | 298              | 6,286       | 68.1           | 22.9              |
> |         | No      | 10     | 11,316.8       | 334              | 8,930       | 68.5           | 20.5              |
> |         | No      | 16     | 14,881.1       | 354              | 11,248      | 69.2           | 19.5              |
> |         | **Yes** | —      | 1,593.3        | 236              | 1,416       | **73.6**       | **31.2**          |
>
> While our experiments focus on two widely used benchmarks—enzyme classification and ligand binding affinity—we agree that a deeper analysis of motif learning and its trade-offs would further strengthen the work. We are currently extending our study to the PPI (Protein–Protein Interaction) dataset; however, training models from scratch makes it challenging to obtain meaningful results within a week. We will update the rebuttal with any results we obtain and include them in the camera-ready version if possible.

---

> > ### Comment · Reviewer_obWc · 2025-08-05
> >
> > Thank you for the response. I think the provided experimental parts have resolved my concerns. For the novelty concerns, despite the authors have provided clarifications, I think the unique contribution of the work is to use the second-motif, and the expressiveness analysis is the direct extension of widely-used theories in the GNN expressiveness. I'd like to raise my score to 3, but I recommend the authors to better clarify the contributions.

---

### Official Review · Reviewer_5Kv4 · 2025-07-02

**Clarity:** 3
**Significance:** 3
**Originality:** 3
**Rating:** 4
**Confidence:** 4

**Summary:**

This paper introduces a multiscale Graph Neural Network (GNN) framework that constructs protein representations using secondary structure motifs annotated by DSSP as building blocks. The model employs a two-stage GNN that first learns local interactions within each motif and then models higher-level relationships between the motifs using geometric orientation features.

**Questions:**

1. Regarding Table 8 in Section E.4, it is unclear if SCHull graphs are still applied to construct the whole protein graph in the ablation studies. Clarifying this would provide stronger evidence for the usefulness of the hierarchical framework.
2. The result for GVP-GNN [5] reported in Table 2 is 68.5%, but it appears to be 65.5% in the ProNet paper ("Learning Hierarchical Protein Representations via Complete 3D Graph Networks") [1]. Could the authors please clarify this discrepancy?
3. Could you provide some support for assumption 4.1?


### References
[1] Z. Zhang, Y. Du, et al. (2023). "Learning Hierarchical Protein Representations via Complete 3D Graph Networks."

[2] S. Wang, et al. (2024). "A Theoretically-Principled Sparse, Connected, and Rigid Graph Representation of Molecules." arXiv preprint arXiv:2404.04021.

[3] Z. Zhang, et al. (2024). "ProtGO: Function-Guided Protein Modeling for Unified Representation Learning."

[4] S. Long, et al. (2023). "Clustering for Protein Representation Learning."

[5] B. Jing, S. Eismann, P. Suriana, R. Townshend, & R. Dror. (2021). "Learning from Protein Structure with Geometric Vector Perceptrons."

**Ethical Concerns:**

["NO or VERY MINOR ethics concerns only"]

**Final Justification:**

Score raised from 3 to 4 after author response.

**Limitations:**

Since SSHG does not consistently perform better than some recent baselines, it would be beneficial to emphasize the simple framework and its computational efficiency, which allow it to achieve comparable results.

**Quality:**

3

**Strengths And Weaknesses:**

### Strengths

1. The paper is clearly written with well-explained assumptions and methodology.
1. The method demonstrates promising, improved results on EC reaction classification and ligand binding affinity benchmarks when compared to state-of-the-art methods like ProNet [1].
1. The adoption of SCHull graphs [2] maintains computational efficiency during training after constructing hierarchical representations for proteins.

**Weaknesses**
1. The technical contribution represents incremental novelty, as the approach primarily integrates established methods—including SCHull graphs [2], DSSP secondary structure prediction, and two-stage message passing—without fundamental innovation.
2. The authors did not include results from ProtGO [3] or a related clustering paper [4], which have reported better EC classification performance than Mamba+SSHG. The work in [3] shares a similar concept of clustering proteins to form a multiscale representation, making a comparison worthwhile.
3. It would be beneficial to benchmark the model on a broader range of protein learning tasks, such as the fold classification and protein-protein interaction tasks discussed in [1], to provide a more comprehensive evaluation of its capabilities. Other tools have provided comprehensive benchmarks for this setting: https://pmc.ncbi.nlm.nih.gov/articles/PMC11213157/, https://proceedings.neurips.cc/paper_files/paper/2023/hash/b6167294ed3d6fc61e11e1592ce5cb77-Abstract-Datasets_and_Benchmarks.html

---

> ### Author Rebuttal · Authors · 2025-07-31
>
> We thank the reviewer for this insightful comment. We would like to clarify the key novelties of our framework and highlight the significant improvements it offers beyond prior work. Below, we address your concerns and questions.
>
> ----
> **Q1. The technical contribution represents incremental novelty, as the approach primarily integrates established methods—including SCHull graphs [2], DSSP secondary structure prediction, and two-stage message passing—without fundamental innovation.**
>
> **Response:**
>
> While our work builds upon established components such as DSSP annotations and SCHull graphs [2], we respectfully disagree that our contribution is merely incremental. Our key innovation lies in the principled **integration of biological hierarchy and geometric structure into a unified multiscale GNN framework**, which to our knowledge, is the first to:
>
> - Use **secondary structure motifs** as hierarchical building blocks to define coarse-grained graph nodes, replacing radius-based heuristics with biologically meaningful segmentation—substantially improving efficiency and scalability.
> - Introduce orientation-aware edges between secondary structures using relative frame encodings (e.g., $g_i^\top g_j$), extending prior definitions based on local neighborhoods of individual residues to capture geometric relationships between structural motifs. This is non-trivial, as existing methods for computing such elements do not apply to structural motifs.
> - Provide formal theoretical guarantees, including expressiveness (Theorem 4.2) and sparsity bounds (Proposition 3.2). This contrasts with prior motif-based models, which often lack such guarantees.
>
> This is not a simple stacking of existing tools, but a biologically aligned architectural design that supports both rigid intra-structural and coarse-grained inter-structural modeling, enabling accurate and interpretable learning of long-range dependencies.
>
> We also emphasize the practical impact of our method:
>
> - Our SSHG framework consistently improves accuracy and efficiency across tasks like enzyme classification and ligand binding affinity, validated on multiple backbones (ProNet, GVP-GNN, Mamba). We have added further ablation studies; key results are summarized below. Notably, on ProNet for EC classification, SSHG reduces the average number of edges from 14,881 to 1,593, resulting in nearly a **2× speedup in training time** per epoch (247s → 140s) and a **90% memory reduction** (17,768 MiB → 1,818 MiB), while slightly improving accuracy (87.0% → 87.2%). This marks a significant breakthrough in balancing efficiency and performance.
>
> **Ligand Binding Affinity (LBA)**
>
> | Model                  | RMSE           ↓     | Pearson        ↑     | Spearman       ↑     |     Ave.Time  ↓    |
> |---|-----|--|--|--|
> | GVP-GNN                | 1.529                | 0.441                | 0.432                |   49
> | GVP-GNN + SSHG     | **1.488**                | **0.512**                | **0.477**                |  **35**
> | | | |
> | Mamba                  | 1.457 ± 0.004        | 0.565 ±  0.003    | 0.554 ±  0.004    | **27**
> | Mamba + SSHG           | **1.399 ± 0.003**        | **0.614 ±  0.003**    | **0.610 ±  0.003**    | 29
>
> **Extended Ablation Study on EC**
> | Model   | +SSHG   | Cutoff | Avg. Num Edges | Time (s/epoch) ↓ | Mem (MiB) ↓ | Test Acc (%) ↑ | Acc/Time (×100) ↑ |
> | - | - | - | - | - | -- | -- | ---- |
> | ProNet  | No      | 4      | 1,034.5        | 138              | 1,290       | 78.1           | 56.6              |
> |         | No      | 6      | 4,755.2        | 165              | 7,760       | 82.1           | 49.8              |
> |         | No      | 8      | 8,013.9        | 185              | 9,580       | 85.6           | 46.3              |
> |         | No      | 10     | 11,316.8       | 210              | 14,548      | 86.4           | 41.1              |
> |         | No      | 16     | 14,881.1       | 247              | 17,768      | 87.0           | 35.2              |
> |         | **Yes** | —      | 1,593.3        | 140              | 1,818       | **87.2**       | **62.3**          |
> |         |         |        |                |                  |             |                |                   |
>
>
> - Compared to SCHull, our hierarchical design significantly reduces reliance on dense radius-based connections, resulting in faster training times and lower memory usage while maintaining or even improving predictive performance. We have now included th following table for a more comprehensive comparison addressing this into the revised paper.
>
> | Model                  | RMSE           ↓     | Pearson        ↑     | Spearman       ↑     |     Ave.Time  ↓    |
> |------|---|-----|-----|---|
> | ProNet-Backbone        | 1.458                | 0.546                | 0.550                | 32.1
> | ProNet-Backbone + SCHull| **1.321**        | **0.581 ± 0.001**         | 0.578    | 34.4
> | ProNet + SSHG| 1.435 ± 0.004    | **0.579 ± 0.004**    | **0.591 ± 0.003**    | **24**
>
>
> ---
>
> **Q2. The authors did not include results from ProtGO [3] or a related clustering paper [4], which have reported better EC classification performance than Mamba+SSHG. The work in [3] shares a similar concept of clustering proteins to form a multiscale representation, making a comparison worthwhile.**
>
> **Response:**
>
> We thank the reviewer for highlighting these important works and will include detailed discussions and comparisons of ProtGO [3] and the clustering method [4] in the revised manuscript. Briefly:
> - ProtGO focuses on Gene Ontology prediction using transformers on full protein sequences, differing from our 3D structure-based EC classification and secondary structure motif modeling.
> - While [4] uses clustering of residues for multiscale representation, we leverage **biologically grounded secondary structure** motifs with provable sparsity and expressiveness guarantees.
> - Although these methods show strong performance, they rely on much larger and more computationally intensive architectures (e.g., ProtGO uses channel sizes up to 2048), whereas our model is significantly more lightweight (channel size up to 256), highlighting efficiency benefits.
>
> We will include this discussion and are happy to add empirical results. However, we found that these models are not yet publicly available. Nevertheless, our work addresses a distinct problem using a compact and efficient architecture that achieves competitive performance. We appreciate the reviewer’s valuable suggestions and will cite these works appropriately.
>
> ---
> **Q3. It would be beneficial to benchmark the model on a broader range of protein learning tasks, such as the fold classification and protein-protein interaction tasks discussed in [1], to provide a more comprehensive evaluation of its capabilities.**
>
> **Response:**
>
> We appreciate the reviewer’s thoughtful suggestion and agree that evaluating across a broader range of protein learning tasks would further demonstrate the generality of our framework. Our current focus on enzyme classification and ligand binding affinity prediction was motivated by their biological relevance and the availability of high-quality benchmark datasets with 3D structural annotations. We are currently working on the Protein–Protein Interaction (PPI) task and training models from scratch, which makes obtaining meaningful results within a week challenging. We plan to include these results in the future and will discuss this ongoing extension in the revised manuscript.
>
> ---
> **Q4. Regarding Table 8 in Section E.4, it is unclear if SCHull graphs are still applied to construct the whole protein graph in the ablation studies. Clarifying this would provide stronger evidence for the usefulness of the hierarchical framework.**
>
> **Response:**
>
> We thank the reviewer for highlighting the need for clarification. In the experiments reported in Table 8, baseline models without our hierarchical framework (SSHG) do not use SCHull graphs; instead,they rely on the default graph construction in these frameworks, which uses radius graphs with a large cutoff. This design clearly isolates the impact of introducing SSHG. Additionally, to directly compare SCHull and SSHG, we have included a new table, as presented in our response to Q1.
>
> ---
> **Q5. The result for GVP-GNN [5] reported in Table 2 is 68.5%, but it appears to be 65.5% in the ProNet paper ("Learning Hierarchical Protein Representations via Complete 3D Graph Networks") [1]. Could the authors please clarify this discrepancy?**
>
> **Response:**
>
> Thank you for catching this important point. As reported in Table 4 (Section 5.3), the baseline GVP-GNN uses a radius cutoff of 4.5 Å, achieving 65.5% accuracy. Increasing the cutoff to 10 Å improves accuracy to 68.5%, as also listed in Table 2 for comparison. We have now clarified this point in the revised paper.
>
> ---
> **Q6. Could you provide some support for assumption 4.1?**
>
> **Response:**
>
> As introduced in the Background section, Assumption 4.1 is necessary for leveraging the theoretical framework of the Weisfeiler-Lehman (WL) test to analyze the maximal expressiveness of GNN architectures—that is, to characterize the best possible representational power an architecture can achieve. This framework is widely adopted across the literature, e.g., [16, 25, 37, 39] (cited in our paper). However, as is common in practice, we do not enforce this assumption strictly. Instead, we follow the standard approach of employing sufficiently expressive multilayer perceptrons (MLPs) with appropriate nonlinearities and sufficient width, as supported by prior works mentioned above. Our implementation adheres to this principle by using MLPs with ReLU activations and sufficient capacity to approximate favorable mappings over the relevant input domains. We have refined the discussion of maximal expressiveness and its underlying assumptions in the revised paper to enhance clarity.

---

> > ### Comment · Reviewer_5Kv4 · 2025-08-06
> > **thank you**
> >
> > The authors have resolved all issues. I have raised my score.

---

> > > ### Author Response · Authors · 2025-08-06
> > > **Thank you**
> > >
> > > Thank you for considering our rebuttal and for your support.

---

### Note · Authors · 2025-08-11

Dear Reviewers and AC,

We thank the reviewers and AC for their constructive feedback throughout the discussion phase. The primary concerns raised were (1) **the novelty of our proposed work** and (2) **the sufficiency of empirical evidence supporting our claims**. According to the response from reviewers to our rebuttal, we believe we have addressed all concerns raised by the reviewers. In particular, the reviewers found the significance of our work:

- “**It is a simple idea, but it is widely applicable across many architectures and seems surprisingly impactful.**” (Reviewer kpTM)


- “**The method demonstrates promising, improved results … maintains computational efficiency.**” (Reviewer 5Kv4)

Moreover, the reviewers acknowledged that our additional experiments and clarifications addressed their earlier concerns:

- “The provided experimental parts have resolved my concerns.” (Reviewer obWc)

- “The authors have resolved all issues.” (Reviewer 5Kv4)

- “I think this has greatly improved your paper and the comparison to other methods is now much clearer.” (Reviewer Ckw3)

Regarding the current rating, we believe that **there are some inconsistencies between Reviewer Ckw3’s positive comment and the negative rating**. We would be grateful if the reviewer could consider updating the score to reflect the positive assessment. Additionally, we believe that **Reviewer obWc’s rating may still stem from some misunderstandings regarding our contributions and novelties**. Reviewer kpTM has pointed out the novelties of our work, and we would like to further clarify the novelty of our work in the following:

**Novelty clarification:** Our work introduces the first multiscale hierarchical framework for protein modeling that directly leverages secondary structure motifs, rather than residue-level graphs, for graph construction. We also provide, to the best of our knowledge, the first provable expressiveness analysis and efficiency for a multiscale framework on geometric data. This theoretical foundation is complemented by a focus on scalability and efficiency—achieving ~90% reduction in memory usage and ~2X training speedup—while maintaining or improving accuracy across multiple backbone architectures.

**Empirical sufficiency:** During rebuttal, we expanded our experiments with additional baselines and ablation studies, demonstrating consistent gains in both accuracy and efficiency.

Thank you so much again for considering our final remarks.

Regards,

Authors

---

### Decision · Program_Chairs · 2025-09-17

**Decision:**

Accept (poster)

**Comment:**

The paper proposes a simple yet effective neural network design for protein representation learning. The method proposes to apply GCN layers on two distinct graphs. The first one is constructed as independent subgraphs of residue groups, grouped by secondary structures, while the second one is constructed by representing each secondary structure as a node and edges between those representing spatial proximity. The paper received initial mixed reviews, with concerns regarding limited evaluation. During the reviewer-author discussion, most of these concerns were addressed, and most of the reviewers provided a positive assessment of the work. After the AC-reviewer discussion, reading the reviews, and the paper, I agree with the overall assessment of the paper and therefore recommend accepting the paper. I encourage the authors to include all the additional baselines and experiments provided during the reviewer-author discussion into the final paper, and the additional protein-protein interaction experiment promised during this period.